# msData: A Millisecond-Resolution Network Dataset for Advancing Time Series Foundation Models

## Abstract

Time series foundation models (TSFMs) require diverse, real-world datasets to adapt across varying domains and temporal frequencies. However, current large-scale datasets predominantly focus on low-frequency time series with sampling intervals, i.e., time resolution, in the range of seconds to years, hindering their ability to capture the nuances of high-frequency time series data. To address this limitation, we introduce a novel dataset, **msData**, that captures millisecond-resolution wireless and traffic conditions from an operational 5G wireless deployment, expanding the scope of TSFMs to incorporate high-frequency data for pre-training. Further, the dataset introduces a new domain, namely, wireless networks, thus complementing existing more general domains like energy and finance. The dataset also provides use cases for short-term forecasting, with prediction horizons spanning from 1 millisecond (1 step) to 96 milliseconds (96 steps). To demonstrate the utility of the dataset, we benchmark traditional machine learning models, transformer-based deep learning models, and TSFMs on forecasting tasks using representative subsets of the data, including a static mobility pattern within YouTube traffic class and a train mobility pattern within Web Browsing traffic class. Across these data distributions, we demonstrate that most TSFM model configurations perform poorly in both zero-shot and fine-tuned settings. Our work underscores the importance of incorporating high-frequency datasets during pre-training and forecasting to enhance architectures, fine-tuning strategies, generalization, and robustness of TSFMs in real-world applications.

## 1 Introduction

Foundation models (FMs) have significantly enhanced machine learning (ML) by utilizing large-scale pre-training on diverse datasets, enabling them to generalize across a wide array of tasks and domains (Thakur, 2024). Recently, time series foundation models (TSFMs) have attracted more interest due to their capability to handle complex temporal tasks, with a particular focus on generalizing across varying time scales and domains, including forecasting, anomaly detection, and classification (Liang et al., 2024). However, developing effective TSFMs requires access to datasets that capture diverse real-world scenarios at varying frequencies and across different domains. The blue dots in Fig.1 demonstrate that the existing benchmark datasets predominantly focus on low-frequency time series with sampling intervals in the range of seconds to years.

Hence, the focus of this paper is to develop and benchmark a high-frequency wireless network dataset in the millisecond resolution (**msData**) and to compare the performance of TSFMs with shallow machine learning models as well as transformer-based deep learning models trained from scratch on this dataset. This dataset enables the study of model behavior in high-frequency settings and potentially provides generalizable and diverse characteristics that can improve the accuracy of TSFMs.

The main contributions of this paper and dataset are: (1) Extending the scope of pre-training and generalizability for state-of-the-art TSFMs by providing a dataset at millisecond resolution (**msData**) (Fig.1). (2) Introduction of a new domain, namely, wireless networks, to the set of existing domains of open datasets (Fig.2). (3) Applications with short-term forecasting, with prediction horizons spanning from 1 millisecond (1 step) to 96 milliseconds (96 steps) (Fig.3).

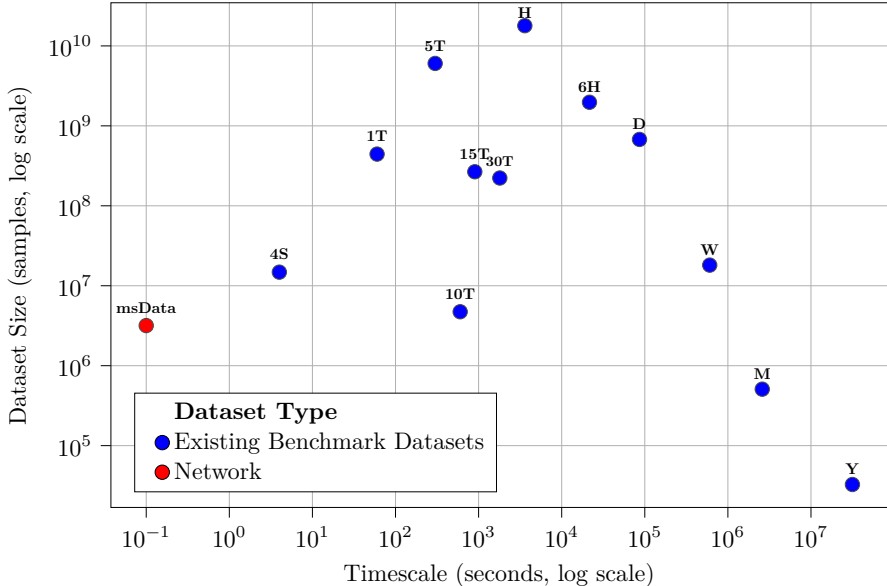

Figure 1: Comparison of timescales and dataset sizes for standard existing datasets used for pre-training (Table 14 in (Aksu et al., 2024)) as compared with the new benchmark. The red dot represents the new dataset that is introduced in this paper.

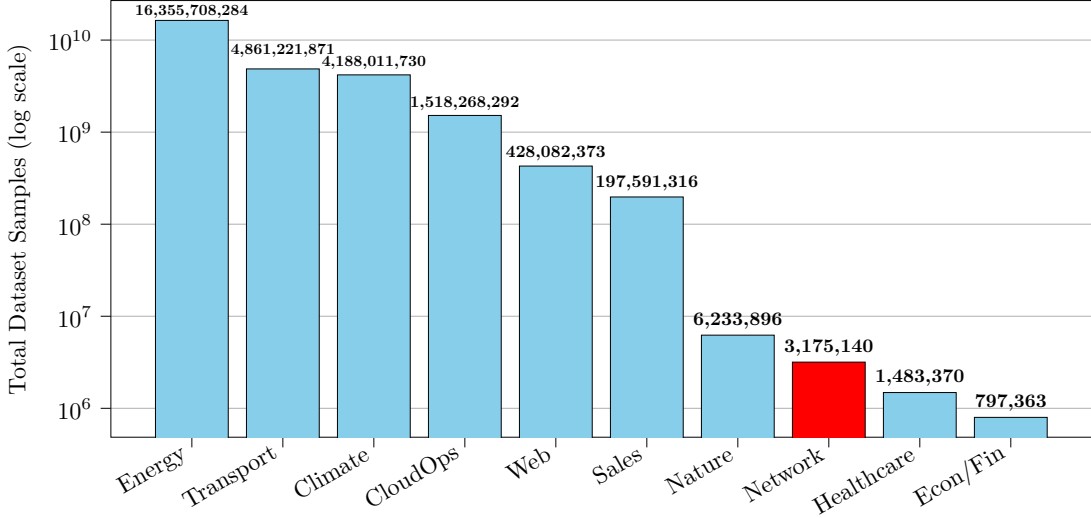

Figure 2: Comparison of existing domains for pre-training (Table 14 in (Aksu et al., 2024)) with the new benchmark. The red bar represents the new dataset that is introduced in this paper.

The rest of the paper is organized as follows. Related work is discussed in Section 2. Section 3 provides a detailed description of the 5G network data, and its characteristics. Section 4 presents the details of models benchmarked, including experimental evaluation and analysis. Section 5 outlines the ablation study. Section 6 provides a detailed performance analysis on an additional filtered data subset. Section 7 discusses the limitations of our work. Finally, in Section 8, we conclude and provide directions for future research.

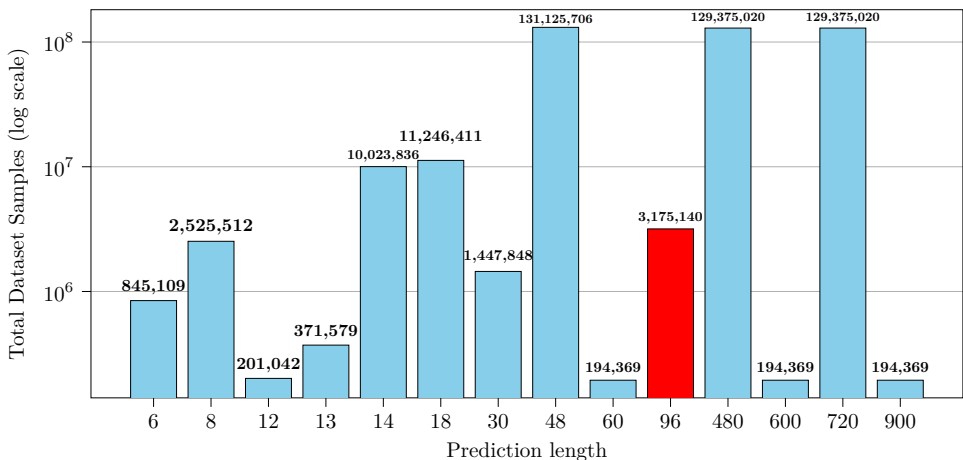

Figure 3: Comparison of prediction lengths of standard test data (Table 2 in (Aksu et al., 2024)) as compared with the new benchmark. The red bar represents the new dataset that is introduced in this paper.

## 2 Related Work

Time Series Foundation Models (TSFMs) have surged in recent years, with their architectures continually evolving to achieve improved performance in both zero-shot and fine-tuned scenarios. Notably, several TSFMs have garnered widespread attention within the community, including Chronos (Ansari et al., 2024), TTM (Ekambaram et al., 2024), Moirai (Woo et al., 2024), TimesFM (Das et al., 2024), and Time-MOE (Xiaoming et al., 2025). These models can be broadly categorized into two distinct classes: transformer-based and non-transformer-based architectures (Liang et al., 2024). Our work complements these developments by introducing a high-frequency, real-world dataset from a novel domain (wireless networks), which provides an additional and challenging benchmark for evaluating the robustness and adaptability of TSFMs.

Transformer-based TSFMs largely follow established self-supervised (e.g., Moirai) or supervised transformer frameworks (e.g., TimeXer), which have garnered significant recognition within the field. In contrast, non-transformer-based TSFMs leverage alternative machine learning models such as Multi-Layer Perceptron (MLP) and Convolutional Neural Networks (CNN) (e.g., TTMs). More recent efforts have also focused on enhancing diffusion-based methods (Kollovieh et al., 2023; Su et al., 2025) for modeling and generating data of different characteristics, which is crucial for generative time series forecasting. Furthermore, to address statistical heterogeneity in time series foundation model training and ensure robust generalization, a decentralized cross-domain model fusion approach, as Federated Learning (FL), has been explored in (Chen et al., 2025).

The successful deployment of these TSFMs for accurate zero-shot forecasting relies on the development of pre-trained models that have undergone extensive training on datasets characterized by diverse patterns and resolution properties. This emphasis on data diversity is critical, as it enables TSFMs to exhibit generalizability across a wide range of scenarios and capture complex temporal dynamics with enhanced accuracy. Notably, prior research has underscored the importance of resolution and domain diversity in pre-trained models for optimizing performance (e.g., Section 4 in (Ansari et al., 2024) for Chronos and Section 4.9 and Fig.3 in (Ekambaram et al., 2024) for TTM.

In practice, a range of open datasets is available for TSFMs, which collectively provide the necessary heterogeneity to ensure that these models generalize effectively to out-of-domain datasets and real-world applications. Specifically, popular datasets such as those from Monash (Godahewa et al., 2021), LIBCITY (Wang et al., 2021), and the UCI Machine Learning archive (Asuncion et al., 2007) have become foundational in pre-training TSFMs and are widely utilized for assessing model performance. These datasets not only serve as data for pre-trained models but also enable out-of-domain testing of pre-trained models when a subset of the datasets are not considered for pre-training. We position our dataset as a complementary resource to these existing open datasets, specifically targeting the gap for millisecond-level time series from communica-

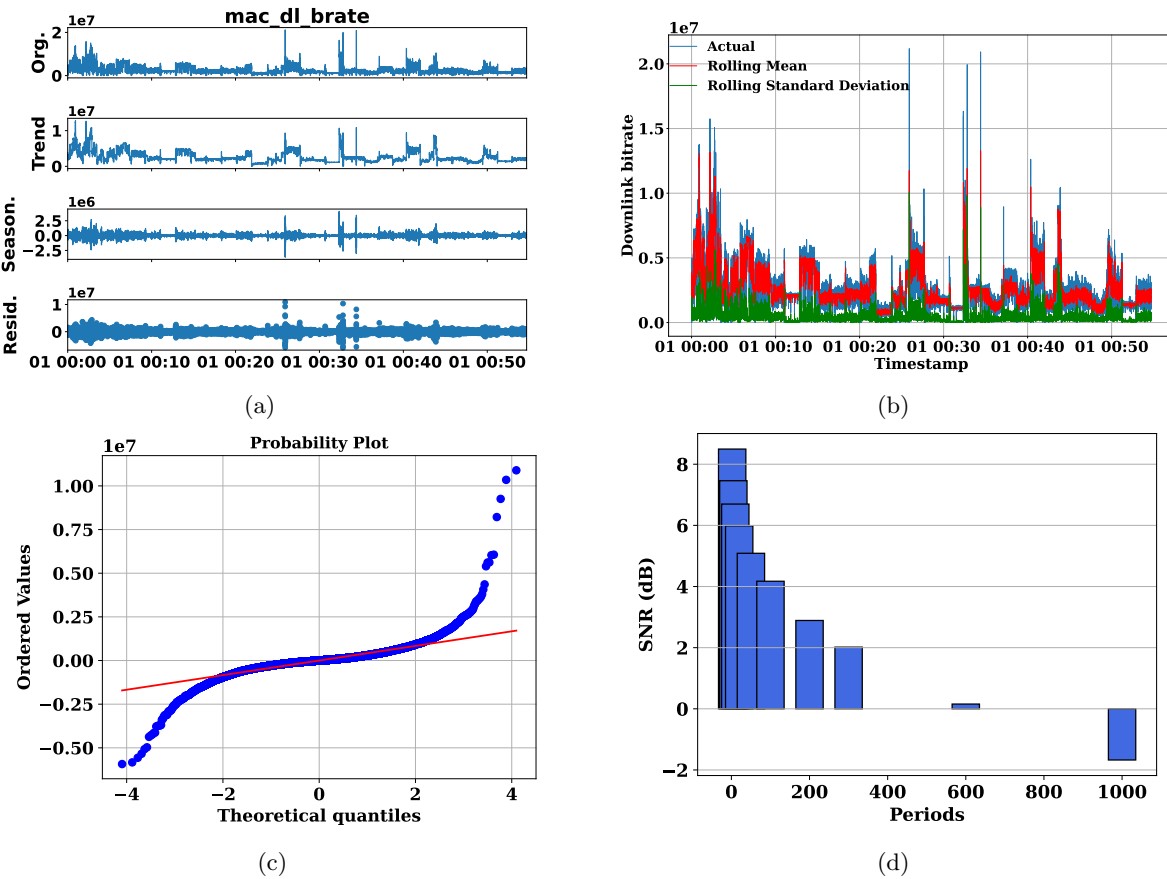

Figure 4: Target variable (Downlink Bitrate; mac_dl_brate): (a) STL decomposition, (b) Rolling mean and standard deviation, (c) Residual Q-Q, (d) Signal-to-Noise Ratio (dB).

tion networks for both training and out-of-domain evaluation of TSFMs. Our dataset directly addresses this need for diversity by introducing a previously underrepresented domain with very fine temporal granularity, thereby contributing to a better understanding of the generalization capabilities of TSFMs when applied to high-frequency wireless data.

This paper introduces a benchmark dataset, **msData**, containing high-frequency time series in wireless networks. It provides curated use cases for univariate and multivariate forecasting and an initial benchmark study on this data.

## 3 Dataset

### 3.1 Dataset Overview

We utilize a time series dataset of 5G Radio Access Network (RAN) Performance Measurements (PMs) collected from a real-world deployment of a 5G Open Radio Access Network (O-RAN) within the OpenIreland testbed. O-RAN introduces a modular and open architecture that decomposes the traditional monolithic RAN into standardized, interoperable components (i.e, the Central Unit (CU), Distributed Unit (DU), and Radio Unit (RU)) facilitating multi-vendor deployments and software-driven control. Central to O-RAN's programmability is the near-Real-Time RAN Intelligent Controller (near-RT RIC), which enables rapid, feedback-driven network optimization.

The data was captured using software-defined radios (Ettus USRPs) configured as a base station and multiple user equipments (UEs). To simulate diverse real-world usage, the setup incorporated various mobility profiles

Table 1: Summary of STL Decomposition of all datasets.

| Dataset | STL Decomposition | | |
|---|---|---|---|
| | **Trend** | **Seasonality** | **Residuals** |
| **Network** | Unstable, step-like shifts. | Weak short-term periodic patterns. Hidden by noise. | Sharp spikes. Bursts of noise. Lots of unpredictable variation. |
| **ETTh1** | Mostly steady with small rises and falls. | Small, regular repeating pattern. | Tiny random changes. |
| **Electricity** | Remains steady throughout. | Strong repeating pattern. | Occasional bursts of noise. |
| **Weather** | Almost flat but interrupted by sudden sharp spikes. | No seasonality. | Mostly small, but with rare sudden jumps. |
| **Traffic** | Slowly increasing trend over time. | Strong, regular repeating pattern. | Small random changes. |

(static, pedestrian, car, bus, and train) and generated traffic from both benign applications (Web Browsing, VoIP, IoT, and video streaming) and malicious activities (DDoS-Ripper, DoS-Hulk, PortScan, Slowloris). PMs were collected at the base station side and span a broad set of physical and medium access control layer features, including the Channel Quality Indicator (CQI), Modulation and Coding Scheme (MCS), Noise ratio interference (SINR), Signal strength (RSSI), buffer occupancy, and packet delivery statistics. In the dataset, each UE is associated with a unique identifier, denoted as *ue_ident*, which serves to distinguish individual UEs across all collected traces. This identifier remains consistent for a given UE, regardless of the mobility pattern or traffic class associated with its traces. The resulting dataset enables temporal modeling of RAN dynamics under realistic operational conditions.

This data context is particularly well-suited for very short-term forecasting, where the goal is to predict network states (e.g., throughput, channel quality, traffic class) over a short horizon ranging from milliseconds to a few seconds. Such forecasting enables predictive control strategies in scenarios characterized by rapid fluctuations in load, mobility, or interference (see Section 3.2 for dataset characteristics). Short-term throughput predictions enhance scheduling efficiency and application-level rate control, especially in latency-sensitive services like cloud gaming or interactive video. Forecasting CQI, for example, allows the network to proactively steer users to cells with better anticipated radio conditions, supports load-aware handovers, and preemptively adjusts adaptive bitrate algorithms for video streaming. Likewise, anticipating traffic class transitions supports early enforcement of QoS policies, dynamic resource allocation (e.g., in network slicing), and intrusion detection mechanisms capable of identifying malicious activity before it significantly degrades the service.

### 3.2 Dataset Characteristics

While Section 3.1 provides a broad overview of the 5G network dataset, our analysis and experiments are carried out on a carefully filtered subset of the data. We filter the raw data on the basis of the mobility pattern and benign traffic class. In particular, we focus on the *static* mobility pattern for the *video streaming* traffic class. Therefore, the results presented here represent the characteristics of the filtered dataset rather than those of the complete dataset.

The time series of the 5G network demonstrates several important characteristics. Fig.4a shows the STL (Seasonal and Trend decomposition using Loess) of the time series, which separates the original data (labeled Org. in Fig.4a) into distinct structural components, i.e., the trend, seasonal, and residual components. Here, the trend component reflects the underlying structure of the series; however, it appears unstable, as characterized by step-like shifts rather than a smooth trajectory. The seasonal component captures only weak short-term periodic patterns, which are easily obfuscated by the stronger irregular behavior in the data. The residual component contains the remaining variability, including sharp spikes and bursts of endogenous noise

that cannot be explained by trend or seasonality. Similarly, as illustrated in Fig.4b, both the rolling mean and the standard deviation are observed to change substantially over time, confirming that the process is non-stationary and heteroskedastic. This means that the statistical properties of the data are not constant. The data exhibit extreme outlier events that are more prominent in specific time periods than in random events throughout the series. The autocorrelation analysis (see Fig.9, Section A.3) reveals a strong temporal persistence with slow decay, confirming the clustering of extreme events observed in the data. In Fig.4c, the residuals deviate strongly from the reference line, particularly in the tails, indicating a heavy-tailed distribution. Finally, the signal-to-noise ratio (SNR) analysis in Fig.4d provides a quantitative view of this instability. The SNR values highlight that the series is dominated by short-term periodic structures (high SNR in periods 2-20), while medium-term cycles exist but are weaker, and long-term seasonality is essentially absent (SNR is near zero and even negative beyond period 600). Overall, the time series is mostly influenced by short-term changes, bursts of volatility and clustered anomalies, rather than stable long-term trends.

Next, we summarize the comparison between our 5G network dataset and other common pre-trained datasets (further experimental details are presented in Appendix A.3). The pre-trained datasets used for comparison are: **ETTh1 (Zhou et al., 2021)** is an hourly subset of the Electricity Transformer Temperature (ETT) dataset, containing two years of transformer oil temperature and related power load data from two counties in China. **Electricity (Wu et al., 2021)** dataset contains the hourly electricity consumption (in kWh) from 321 clients, recorded between 2012 and 2014. **Weather (Wu et al., 2021)** data from 2020 in Germany, recorded every 10 minutes, with 21 indicators such as air temperature, humidity, and wind speed. **Traffic (Wu et al., 2021)** is a collection of hourly road occupancy rates (0–1) from sensors on San Francisco Bay Area freeways, collected by the California Department of Transportation between 2015 and 2016. Table 1 summarizes the key differences among the datasets based on their STL decomposition, highlighting that our dataset is notably different due to its unstable trend, weak seasonality, and spiky residuals. Appendix A.3 includes other data characteristics, such as temporal dependencies, and statistical variability.

## 4 Benchmark

In this section, we provide a comprehensive analysis of the benchmarked models (as explained in 4.1) for the considered target variable *downlink bitrate* (*bitrate*) in the 5G network dataset. In the multivariate setting, all considered models use four input features, with descriptions provided in Table 2. Section 4.3 provides implementation details, including the data processing pipeline, that reflects our consideration of only a subset of data to illustrate the impact of this high-frequency dataset.

Table 2: Features used in multivariate setting.

| Feature | Description |
| --- | --- |
| CQI | Channel Quality Indicator |
| MCS | Modulation and Coding Scheme |
| pkt ok | Number of packets sent |
| pkt nok | Number of packets dropped |

### 4.1 Models benchmarked

We selected three state-of-the-art tree-based ensemble models: Random Forest (RF) (Breiman, 2001), implemented using Scikit-learn, eXtreme Gradient Boosting (XGBoost, hereafter XGB) (Chen & Guestrin, 2016), and Adaptive Random Forest (ARF) (Gomes et al., 2018) implemented using the River library. As an additional online baseline, we included a simple incremental linear regression model, Online LR (OLR) (Ouhamma et al., 2021), also implemented using the River library. Similarly, we selected a non-parametric baseline, referred to as naive forecast (Naive) (Beck et al., 2025), for a fair evaluation on high-frequency data.

As a deep learning forecasting baseline, we additionally evaluated PatchTST (Nie et al., 2022) and iTransformer (Liu et al., 2024). PatchTST is a transformer-based model that segments time series into patches and

Table 3: Parameters used in model training.

| Parameter | Univariate | Multivariate |
|---|---|---|
| n_models | 10 | 20 |
| max_features | None | 0.5 |
| grace_period | 50 | 100 |
| max_depth | None | 5 |

(a) Hyper-parameters specific to ARF.

| Parameter | Value |
|---|---|
| Target variable | Downlink bitrate |
| No. of features | 4 |
| Mobility pattern | Static |
| Past observations | 5 |
| Prediction horizon | 96 |
| Train set:Test set | 80:20 |

(b) Common parameters for all shallow models.

applies channel-independent encoding for efficient long-term forecasting. iTransformer is an inversion-based transformer architecture that treats variates as tokens to better capture multivariate correlations in time series forecasting.

In addition, we evaluated three time series foundation models (TSFMs): TinyTimeMixer (TTM) (Ekambaram et al., 2024), Chronos (Ansari et al., 2024), and Lag-Llama (Rasul et al., 2023), each specifically designed for time series forecasting. TTM is an extremely light-weight pre-trained model, with effective transfer learning capabilities based on the light-weight TSMixer architecture. Likewise, Chronos is a language modeling framework for time series for pre-trained probabilistic time series models. In this work, we specifically adopted the Chronos-bolt-small variant (46M parameters) as the representative Chronos model for our experiments. Lag-Llama is a general-purpose foundation model for univariate probabilistic time series forecasting based on a decoder-only transformer architecture that uses lags as covariates.

### 4.2 System specification

The experiments are carried out on a local machine with the following hardware and software specifications: **Operating System:** Microsoft Windows 10 Enterprise, Version 22H2; **Processor:** 11th Gen Intel® Core™ i7-1165G7 CPU @ 2.80 GHz with 4 cores and 8 threads; **Memory:** 32 GB RAM.

### 4.3 Implementation details

*Pre-processing:* For shallow models, input sequences are constructed using a sliding-window approach, where past observations within a fixed window are used to predict future target values. For PatchTST and iTransformer, we followed the supervised learning setup provided in the official implementation. For TSFMs, we followed the original implementation protocols described in their respective papers.

*Model parameters:* Table 3 summarizes the parameters used during model training. For common parameters shared across all shallow models, offline experiments were conducted to select optimal values based on prediction accuracy, ensuring fair benchmarking conditions. Both RF and XGB models used these optimized common parameters along with their respective default model-specific hyper-parameters without additional tuning. For the ARF univariate model, we used the default parameters along with the optimized common parameters. For the multivariate setting, model-specific hyper-parameter tuning was performed using random search, with parameter ranges detailed in Table 3a. The best-performing Root Mean Square Error (RMSE)-based ARF configuration was selected for the final evaluation. For PatchTST and iTransformer, since these models are trained from scratch, we adopt a lightweight configuration by setting the embedding dimension to $d_{\text{model}} = 64$ and the feed-forward dimension to $d_{\text{ff}} = 2 \cdot d_{\text{model}} = 128$, while keeping all other hyper-parameters at their default values.

Furthermore, the prediction horizons range from 1 millisecond up to 96 milliseconds. Short-term horizons are often straightforward, as the target variable (i.e., bitrate) tends to remain stationary across very small timescales. In contrast, longer horizons provide more meaningful insights, enabling applications such as video streaming to anticipate changes in bitrate and proactively adjust parameters like encoding level. These long horizon forecasts are valuable both for adapting Quality of Service (QoS) and for estimating the stability

Table 4: Performance metrics of benchmarked models.

| Model | Univariate | | Multivariate | |
| --- | --- | --- | --- | --- |
| | RMSE | MAE | RMSE | MAE |
| RF | $0.0344 \pm 0.0001$ | $0.0227 \pm 0.0001$ | $0.0342 \pm 0.0001$ | $0.0226 \pm 0.0001$ |
| XGB | $0.0354 \pm 0.0001$ | $0.0232 \pm 0.0001$ | $0.0354 \pm 0.0001$ | $0.0231 \pm 0.0001$ |
| ARF | $\mathbf{0.0270 \pm 0.0002}$ | $0.0189 \pm 0.0001$ | $\mathbf{0.0175 \pm 0.0007}$ | $\mathbf{0.0130 \pm 0.0005}$ |
| Naive | $0.0418 \pm 0.0000$ | $0.0240 \pm 0.0000$ | $0.0418 \pm 0.0000$ | $0.0240 \pm 0.0000$ |
| OLR | $0.0551 \pm 0.0000$ | $0.0308 \pm 0.0000$ | $0.0555 \pm 0.0000$ | $0.0310 \pm 0.0000$ |
| PatchTST | $0.0327 \pm 0.00044$ | $0.0212 \pm 0.00035$ | $0.0321 \pm 0.00029$ | $0.0207 \pm 0.00009$ |
| iTransformer | $0.0325 \pm 0.00013$ | $0.0208 \pm 0.00019$ | $0.0324 \pm 0.00015$ | $0.0208 \pm 0.00006$ |
| TTM (Zero-shot) | $0.0359 \pm 0.0000$ | $0.0230 \pm 0.0000$ | $0.0359 \pm 0.0000$ | $0.0230 \pm 0.0000$ |
| TTM (Fine-tuning) | $0.0371 \pm 0.0015$ | $0.0237 \pm 0.0011$ | $0.0393 \pm 0.0007$ | $0.0250 \pm 0.0004$ |
| Chronos (Zero-shot) | $0.0313 \pm 0.0000$ | $0.0185 \pm 0.0000$ | $0.0273 \pm 0.0000$ | $0.0181 \pm 0.0000$ |
| Chronos (Fine-tuning) | $0.0281 \pm 0.0000$ | $\mathbf{0.0178 \pm 0.0000}$ | $0.0253 \pm 0.0000$ | $0.0176 \pm 0.0000$ |
| Lag-Llama (Zero-shot) | $0.0617 \pm 0.0002$ | $0.0384 \pm 0.0001$ | - | - |
| Lag-Llama (Fine-tuning) | $0.0474 \pm 0.0039$ | $0.0268 \pm 0.0009$ | - | - |

of the bitrate, that is, how frequently it is expected to change. The dataset is divided into 80% training and 20% testing, preserving temporal order. As the data for each user are sequential and not mixed, this split naturally keeps the sequence of each user intact, preventing data leakage from future observations into training. However, for PatchTST and iTransformer, we adopt a 70%/10%/20% split for training, validation, and testing, respectively.

*Model training:* For RF and XGB, we utilized Scikit-learn's *MultiOutputRegressor* wrapper to enable direct multi-step forecasting. In the case of PatchTST, we followed the original supervised forecasting protocol described in the paper and official implementation. Specifically, the historical input sequence was partitioned into fixed-length temporal patches, and multivariate forecasting was performed using a channel-independent Transformer encoder with shared weights across variables. Similarly, for iTransformer, we followed the original forecasting protocol described in the paper and official implementation. Specifically, we employed an inverted transformer design that treats individual variates as tokens, allowing the self-attention mechanism to capture inter-variate dependencies more effectively for multivariate time series forecasting. For TSFMs, we followed the original implementation protocols described in their respective papers.

*Post-processing:* Both Chronos and Lag-Llama are trained to predict a fixed length horizon **H** from a given context window. By default, these models produce forecasts only for the final prediction window of each series and skip series that do not meet the minimum context length. This default evaluation framework differs from shallow models that generate forecasts for every test sample. To ensure a consistent comparison across models, we implemented a rolling evaluation procedure for both Chronos and Lag-Llama. Specifically, starting with each timestamp **t**, we provide the model with all historical data available up to **t** and generate the next steps **H**. We then slide the starting point forward by one time step and repeat the prediction until the end of the series. This produces overlapping multi-step forecasts aligned with each test timestamp, allowing direct comparison with the shallow models.

In the multivariate setting, the covariates CQI, MCS, pkt_ok, and pkt_nok are treated as synchronously observed features rather than future unknowns. This assumption is not an idealization: in an operational O-RAN deployment, the Distributed Unit (DU) reports these Physical and Medium Access Control (PHY/MAC) layer performance measurements to the near-Real-Time RAN Intelligent Controller (near-RT RIC) via the E2 interface at configurable periodicities in the sub-second range, consistent with the millisecond resolution of our dataset. Consequently, at each prediction timestep **t**, all covariate values corresponding to that timestep are already observed quantities delivered through standard O-RAN measurement reporting, and no future covariate leakage occurs in our experimental setup.

To extend Chronos, which is inherently a univariate model, to the multivariate setting, we use AutoGluon-TimeSeries (AG-TS) covariate regressors (Shchur et al., 2023). The covariate regressor is a tabular model trained on known covariates and static features to predict the target at each time step. Its predictions are subtracted from the target series, and Chronos then forecasts the residuals. For each rolling window, we create a future covariate table that matches the next **H** time steps immediately following the end of the current window. This table contains the values of the exogenous variables for those steps, allowing Chronos to use both the historical target and the future covariates to generate accurate forecasts.

### 4.4 Results

In this section, we evaluate the performance of shallow models, transformer-based deep learning models, and TSFMs in both univariate and multivariate settings. Table 4 presents the performance of the benchmarked shallow models and TSFMs, evaluated using RMSE and Mean Absolute Error (MAE). Both metrics (RMSE and MAE) are computed on the normalized target (downlink bitrate) values. Each model was trained with three different random seeds (42, 99, and 123) to ensure reproducibility and to assess variability in

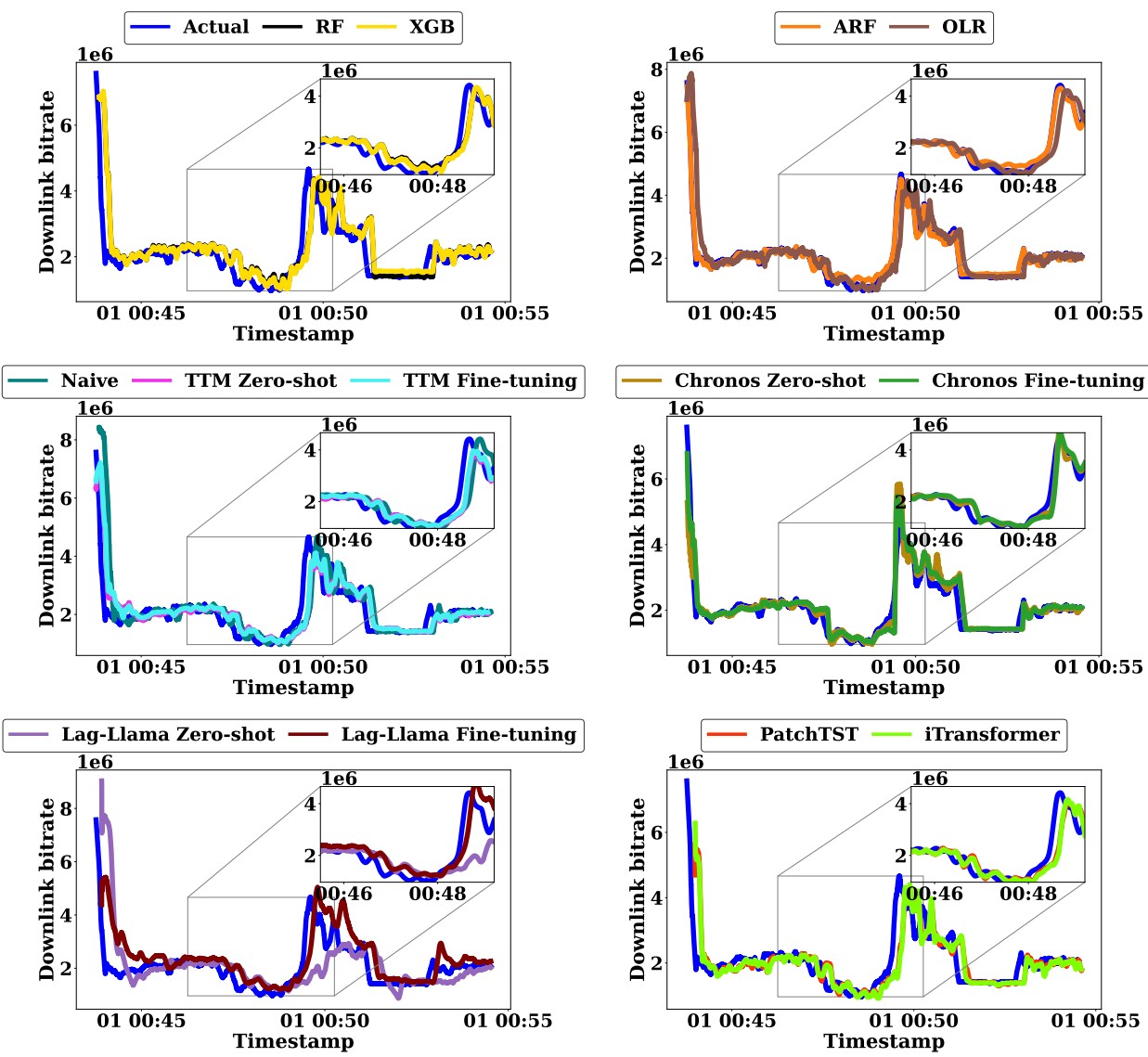

Figure 5: Actual v.s. Predicted bitrate values in a Univariate setting.

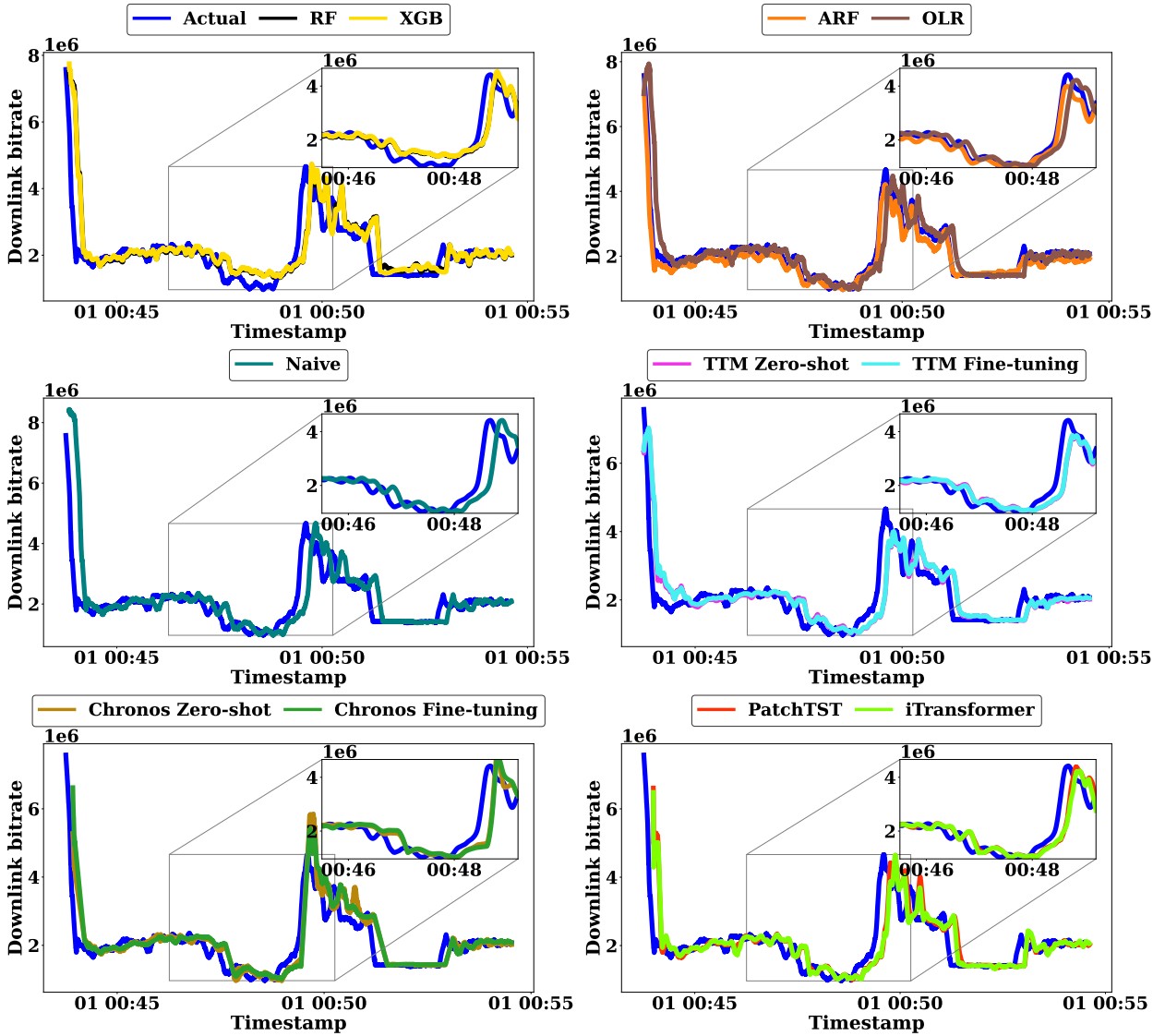

Figure 6: Actual v.s. Predicted bitrate values in a multivariate setting.

performance. ARF achieves the best performance among the evaluated models across both settings. The performance gain is consistent with the data characteristics observed in Section 3.2; our msData is dominated by irregular spikes, step-like changes, and lack of stable seasonality. Static models such as RF or XGB struggle in performance because they assume that the training distribution does not change over time, leading to poor generalization when sudden data shifts occur. Although the Online LR baseline is updated incrementally, it still cannot fully capture the complex, non-linear dynamics in the data. Similarly, PatchTST and iTransformer underperform compared to ARF, even when trained from scratch and extensively tuned over key hyper-parameters (see Section A.4.4). The hyper-parameter tuning results show limited improvements on this dataset.

Moreover, TSFMs performance degrades due to a shift in data distribution in the zero-shot scenario, as these pre-trained models are trained primarily on low-frequency data, limiting their ability to capture high-frequency dynamics with unpredictable spikes and irregular patterns. Based on the results, it can be seen that even after fine-tuning and further hyper-parameter tuning (see Section A.4.3) on our dataset, the performance of TSFMs remains suboptimal, as they fail to generalize effectively. In contrast, ARF is designed to handle concept drift by dynamically updating its ensemble of trees as new patterns appear. This allows it to

Table 5: Performance Metrics for Different Fine-Tuning strategies for TTM.

| Fine-tuning Strategy | RMSE | MAE |
|---|---|---|
| Head-only fine-tuning | 0.0413 | 0.0270 |
| Adapter-based fine-tuning | 0.0522 | 0.0334 |

Table 6: Performance metrics of benchmarked models with the increasing temporal resolution.

| Temporal Resolution | Prediction Horizon | ARF | | TTM Zero-shot | | TTM Fine-tuning | | Chronos Zero-shot | |
|---|---|---|---|---|---|---|---|---|---|
| | | RMSE | MAE | RMSE | MAE | RMSE | MAE | RMSE | MAE |
| 100 ms | 96 | **0.0457** | **0.0262** | 0.0765 | 0.0434 | 0.0743 | 0.0421 | 0.0622 | 0.0338 |
| 200 ms | 48 | **0.0471** | **0.0267** | 0.0870 | 0.0499 | 0.0880 | 0.0496 | 0.0740 | 0.0389 |
| 500 ms | 20 | **0.0398** | **0.0218** | 0.0855 | 0.0490 | 0.0894 | 0.0542 | 0.0711 | 0.0372 |
| 1000 ms | 10 | **0.0297** | **0.0176** | 0.0856 | 0.0500 | 0.0856 | 0.0500 | 0.0580 | 0.0326 |
| 2000 ms | 5 | **0.0289** | **0.0169** | 0.0880 | 0.0527 | 0.0915 | 0.0584 | 0.0671 | 0.0354 |
| 3000 ms | 4 | **0.0289** | **0.0185** | 0.1049 | 0.0618 | 0.1061 | 0.0638 | 0.0860 | 0.0443 |

quickly adapt to data distribution changes and maintain predictive accuracy even in the presence of strong irregularities. While it is observed that Chronos offers a competitive performance in the univariate setting, ARF outperforms Chronos in the multivariate setting.

The performance of these models is more clearly reflected in Fig.5 and Fig.6. We observe that ARF follows the curve/trend of the **bitrate** much better than the other shallow models, transformer-based models, and TSFMs. For the purpose of visualization, we average the actual and predicted values for each test sample.

## 5 Ablation Study

### 5.1 Fine-tuning Strategies for TTM

In this section, we analyze how different fine-tuning strategies affect the performance of TTM. We explore two different fine-tuning strategies: **(i) Head-only fine-tuning** (Ekambaram et al., 2024), where we freeze the entire backbone and decoder and only train the final prediction head, **(ii) Adapter-based fine-tuning** (Houlsby et al., 2019), where we incorporate lightweight MLP adapter modules inside the mixer blocks while keeping the original TTM weights frozen. Recent works on fine-tuning TSFMs (Tomar et al.) has shown that even widely used Parameter-Efficient Fine-Tuning (PEFT) methods like Low-Rank Adaptation (LoRA) do not consistently improve the performance of TSFMs. Our findings in Table 5 align with this observation; even though both the fine-tuning strategies are architecturally compatible with TTM, their performance is worse than the default TTM fine-tuning approach.

### 5.2 Temporal Resolution

In this section, we evaluate the performance of ARF and TSFMs in a multivariate setting. We analyze the effect of increasing the temporal resolution of the data on the performance of both ARF and TSFMs, performing fine-tuning only for TTM because of its computational efficiency. The prediction horizon is fixed at 9.6 seconds (96 steps) for all temporal resolutions. Specifically, we evaluate these models on newly filtered data: the **pedestrian** mobility pattern for the **video streaming** traffic class, to highlight that the characteristics of our dataset differ from those of the pre-trained datasets. Table 6 shows the performance of ARF and TSFMs. Notably, increasing the temporal resolution does not improve the performance of TSFMs. In contrast, ARF consistently outperforms TSFMs at each resolution, as higher temporal resolution reduces noise and improves its predictions. This indicates that TSFMs perform poorly not only because of temporal resolution (i.e., high frequency), but also due to the inherent characteristics of our data.

Table 7: Performance metrics of benchmarked models on the filtered subset defined by the *train* mobility pattern within the *Web Browsing* traffic class.

| | Univariate | | Multivariate | |
| Model | RMSE | MAE | RMSE | MAE |
|---|---|---|---|---|
| RF | 0.0751 | 0.0542 | 0.0740 | 0.0536 |
| XGB | 0.0774 | 0.0536 | 0.0758 | 0.0533 |
| ARF | **0.0605** | **0.0366** | **0.0459** | **0.0230** |
| Naive | 0.0727 | 0.0476 | 0.0727 | 0.0476 |
| OLR | 0.0720 | 0.0470 | 0.0723 | 0.0468 |
| PatchTST | 0.0705 | 0.0468 | 0.0739 | 0.0484 |
| iTransformer | 0.0707 | 0.0453 | 0.0726 | 0.0465 |
| TTM (Zero-shot) | 0.0718 | 0.0460 | 0.0718 | 0.0460 |
| TTM (Fine-tuning) | 0.0719 | 0.0475 | 0.0720 | 0.0488 |
| Chronos (Zero-shot) | 0.0724 | 0.0376 | 0.0886 | 0.0473 |
| Chronos (Fine-tuning) | 0.0694 | 0.0389 | 0.0846 | 0.0470 |
| Lag-Llama (Zero-shot) | 0.0818 | 0.0508 | - | - |
| Lag-Llama (Fine-tuning) | 0.0771 | 0.0460 | - | - |

## 6 Performance Analysis on an Additional Filtered Data Subset

In this section, we evaluate the performance of the benchmarked models on a filtered subset of the dataset characterized by mobility patterns and traffic classes different from those presented in Section 3.2 and 5.2. The raw data is filtered based on mobility patterns and traffic generated from *benign applications*, with a particular focus on the *train* mobility pattern within the *Web Browsing* traffic class. While we focus on this subset here, additional evaluations across different mobility patterns and traffic classes are presented in Appendix A.4.2.

Shallow models, Chronos, and Lag-Llama are evaluated using their default hyper-parameters. For TTM, we use the default hyper-parameters in the zero-shot setting. For fine-tuning, we follow the default configuration, including the built-in learning-rate finder, with the fine-tuning percentage set to 10%. For PatchTST and iTransformer, since these models are trained from scratch, we adopt a lightweight configuration by setting the embedding dimension to $d_{model} = 64$ and the feed-forward dimension to $d_{ff} = 2 \cdot d_{model} = 128$, while keeping all other hyper-parameters at their default values.

Table 7 presents the performance of the benchmarked shallow models and TSFMs. For this filtered subset as well, ARF consistently outperforms other shallow models, transformer-based deep learning models, and TSFMs in both univariate and multivariate settings.

## 7 Limitations

Our current study provides valuable insights into the performance of shallow models, transformer-based deep learning models, and TSFMs for millisecond resolution wireless network data, and shows the need to utilize this dataset to enhance the generalizability and applicability of TSFM pre-training and fine-tuning capabilities. However, there are certain limitations in the study that highlight areas for potential improvement in future research. These include:

- The experiments were conducted on only a limited set of filtered dataset configurations, focusing on a static mobility pattern within the YouTube traffic class and a train mobility pattern within the Web Browsing traffic class. As a result, the findings may not fully generalize across other traffic classes, mobility profiles, or broader wireless network scenarios.

- Further, default implementations of the TSFMs were considered for the performance on zero-shot models. Feature engineering and data pre-processing strategies can potentially improve the performance of TSFMs but this was not considered. Since shallow models work directly on the raw data and perform reliable forecasting, the same was done for TSFMs to make the comparison fair.

- Default fine-tuning implementations were explored for each TSFM, whereas TTM was further evaluated using distinct fine-tuning strategies. However, novel techniques such as autotuning and Low-Rank Adaptation (LoRA) (Hu et al., 2022) strategies were not considered since the focus was on zero-shot and few-shot learning. Future work on ablation studies is proposed to investigate whether optimizing few-shot learning parameters can significantly enhance the performance of TSFMs.

## 8  Conclusion and Future Work

We present a novel high-frequency time series dataset, **msData**, capturing millisecond-resolution measurements from a real-world wireless network. This dataset addresses an important gap in existing large-scale resources, which largely lack fine-grained wireless network data. Our experiments show limitations of current TSFMs in this high-frequency setting and suggest that incorporating diverse, high-resolution datasets during pre-training can be beneficial for improving robustness and generalization. We specifically considered a static mobility pattern for the YouTube traffic class and a train mobility pattern for the Web Browsing traffic class, thereby focusing on two distinct wireless scenarios. In future work, we plan to extend the analysis to additional traffic classes and mobility profiles, as well as explore applications of this dataset in anomaly detection and transfer learning across different mobility conditions.

## Broader Impact Statement

The dataset introduced in this work was collected entirely within a controlled testbed environment using software-defined radios and scripted user equipment, with no participation of real subscribers or transmission of user-generated content. All device identifiers are internal testbed labels assigned to specific hardware units and carry no personally identifiable information. Consequently, the dataset raises no privacy or anonymization concerns of the kind associated with operational network deployments.

The inclusion of malicious traffic classes, namely DDoS-Ripper, DoS-Hulk, PortScan, and Slowloris, is intended to support intrusion detection and traffic classification research within the O-RAN security domain. The data released consists exclusively of aggregated PHY and MAC layer performance measurements collected at the base station side, such as CQI, MCS, SINR, and packet delivery statistics. These are network-side observability metrics and do not expose packet content, application payloads, or any information that would provide meaningful operational uplift to a prospective attacker. The value of including these classes lies in enabling models that can identify anomalous RAN behavior from standard KPIs alone, without reliance on deep packet inspection, which is a defensive rather than offensive capability.

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

# A  Appendix

## A.1  Use cases

The dataset enables a wide range of learning tasks that support more adaptive Radio Access Network (RAN) behavior, particularly within O-RAN systems where short-term predictions and rapid classifications can guide near-real-time control. Its millisecond-resolution measurements, combined with detailed PHY- and MAC-layer indicators and labels for traffic type and mobility class, allow the design of regression models that forecast throughput, channel quality, and link reliability over horizons from a few milliseconds to several seconds. Such predictions can inform scheduling decisions at the DU, guide proactive MCS and power adjustments, improve rate control for latency-sensitive applications, and support mobility steering by anticipating future channel degradation for fast-moving users. The temporal characteristics of the dataset, including irregular bursts and heavy-tailed dynamics, make it well suited for evaluating predictive approaches in environments where rapid fluctuations dominate.

The dataset also supports classification tasks involving mobility identification and traffic-type recognition. Since user movement patterns such as static, pedestrian, car, bus, and train produce distinct combinations of SINR, CQI, and bitrate variability, models trained on these traces can infer mobility behavior directly from RAN KPIs. Such inferences allow the RAN Intelligent Controller (RIC) to select mobility-aware handover strategies, tune power control settings, or commit resources more efficiently. Traffic classification, which extends across benign and malicious flows, provides an additional line of evidence for service-awareness and security monitoring. The dataset includes benign web, VoIP, IoT, and video traffic, as well as multiple attack types such as DDoS-Ripper, DoS-Hulk, PortScan, and Slowloris. This makes it possible to detect abnormal

traffic solely from network-side performance indicators, enabling security functions that do not rely on deep packet inspection.

Beyond supervised learning, the dataset's sharp spikes, volatility clusters, and inconsistent seasonal structure create strong opportunities for anomaly detection. Deviations in CQI, SINR, buffer occupancy, packet loss, or bitrate can reveal early signs of congestion, or malicious activity. Because the dataset includes both dynamic mobility patterns and diverse traffic sources, anomaly detectors built on it can be tested against conditions where network behavior changes rapidly and non-linearly. This setting mirrors real operational networks more closely than traditional low-frequency datasets and supports the design of proactive mitigation strategies within the RIC.

Finally, the dataset's combination of high-frequency time series, labelled mobility classes, and labelled traffic classes allows for multi-task learning and transfer learning studies. Models can be trained on one mobility class and evaluated on another, or jointly predict throughput while classifying user behavior. This supports research on generalization across heterogeneous RAN conditions and offers a realistic foundation for developing predictive, adaptive, and security-oriented control functions that operate within the O-RAN architecture.

## A.2 Performance Evaluation Metrics

The Root Mean Squared Error (RMSE), and Mean Absolute Error (MAE) are calculated as follows:

$$RMSE(Y_t, \hat{Y}_t) = \sqrt{\frac{1}{T} \sum_{t=1}^{T} (Y_t - \hat{Y}_t)^2}, \tag{1}$$

$$\text{MAE}(Y_t, \hat{Y}_t) = \frac{1}{T} \sum_{t=1}^{T} \left| Y_t - \hat{Y}_t \right|, \tag{2}$$

where $Y_t$ and $\hat{Y}_t$ are the actual and predicted bitrate values, and $T$ is the total number of samples in the test data.

## A.3 Data Characteristics Comparison

In this section, we compare our 5G network dataset with those used in the pre-training of TSFMs. The comparison focuses on key data characteristics, including statistical distributions, temporal dependencies, and statistical variability, as illustrated in Figs.7, 8, 9, and 10. We compare the datasets using STL decomposition, rolling mean and standard deviation, autocorrelation (ACF), and residual QQ plots. Our 5G network data is clearly the most different; its trend shifts abruptly in steps, seasonality is weak and mostly hidden by noise, rolling statistics change suddenly, the ACF shows strong temporal persistence with slow decay, and the residual QQ plot departs strongly from normality due to sharp spikes. In contrast, the ETTh1 dataset has a mostly steady trend with mild rises and falls, small but regular seasonal cycles, stable rolling statistics, weak cyclical autocorrelation, and residuals close to normal. The Electricity dataset also remains steady in its trend but shows stronger repeating seasonal patterns, its rolling mean is flat and variance is stable, clear cycles in the ACF, and residuals with occasional deviations. The Weather dataset is mostly flat with rare sharp jumps, no meaningful seasonality, sudden variance spikes in rolling statistics, weak ACF signals, and QQ plots highlighting outliers. Finally, Traffic dataset combines a smooth upward trend with strong, consistent seasonality, gradually increasing rolling mean with stable variance, clear seasonal autocorrelation, and residuals that follow normality fairly well. To conclude, our dataset differs from the others because its persistence comes from clustered extremes and abrupt shifts rather than smooth or cyclical structure, making it the least regular and most unpredictable series.

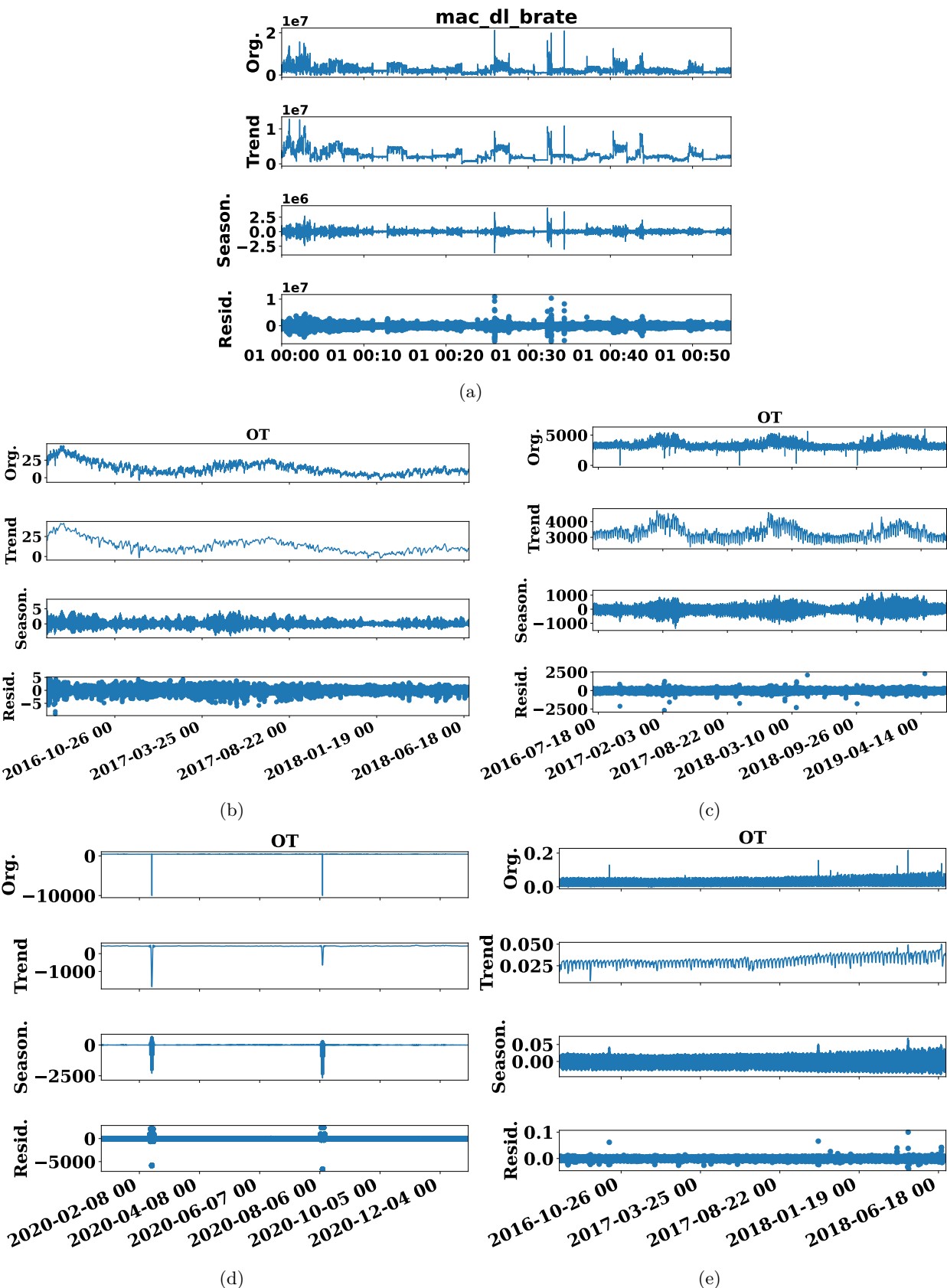

Figure 7: STL decomposition of time series: (a) Network, (b) ETTh1, (c) Electricity, (d) Weather, (e) Traffic.

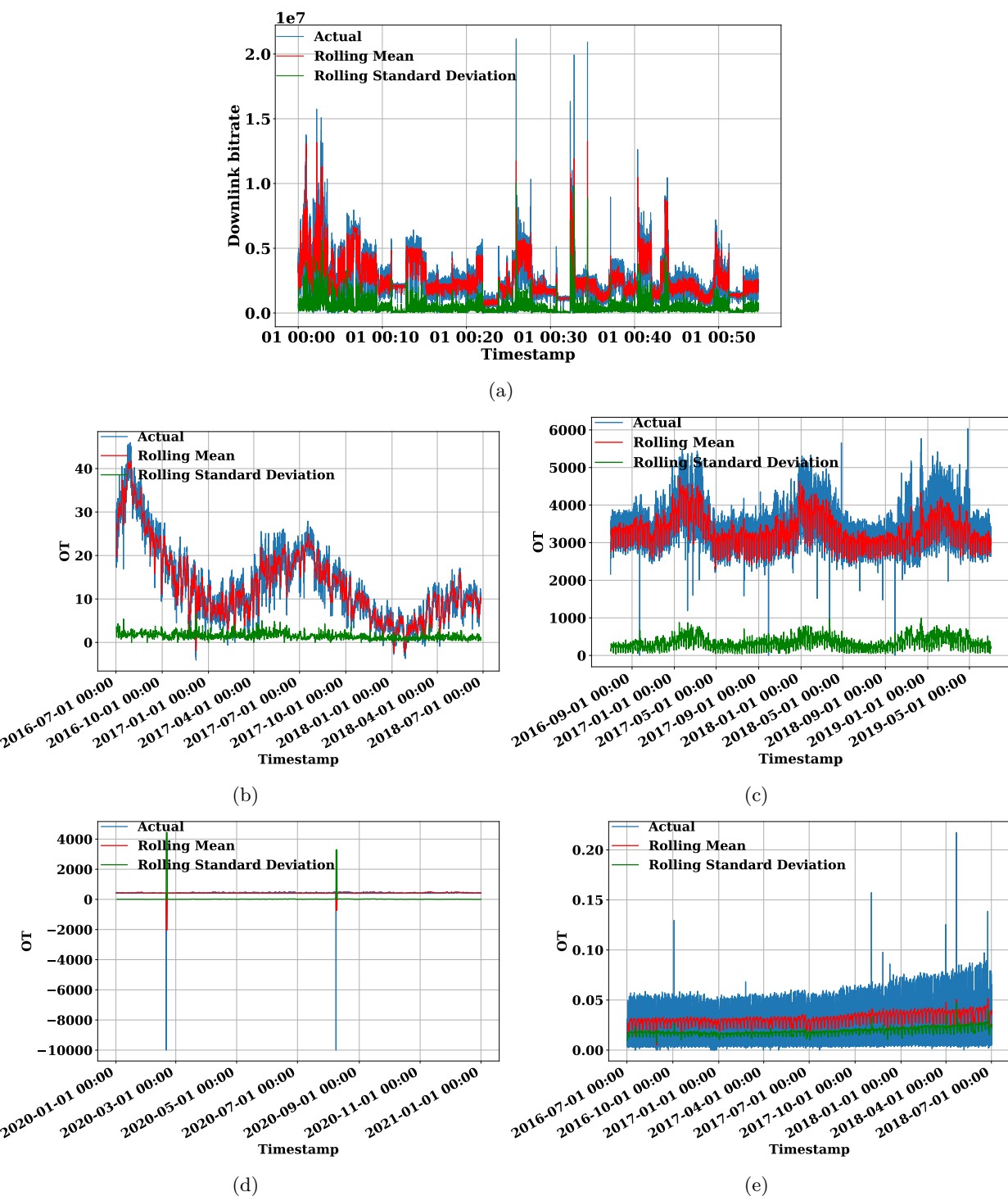

Figure 8: Rolling mean and standard deviation of time series: (a) Network, (b) ETTh1, (c) Electricity, (d) Weather, (e) Traffic.

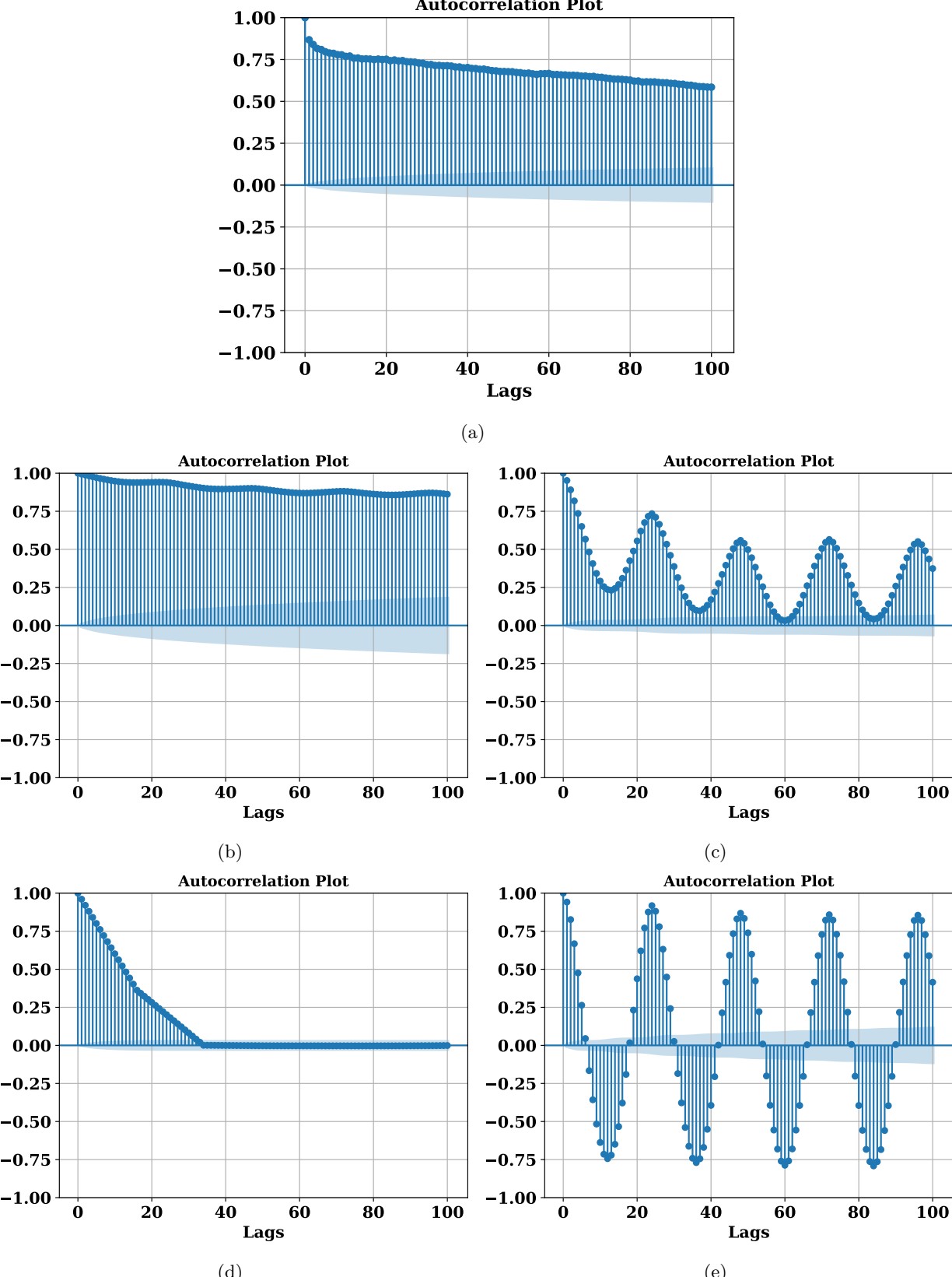

Figure 9: Autocorrelation of time series: (a) Network, (b) ETTh1, (c) Electricity, (d) Weather, (e) Traffic.

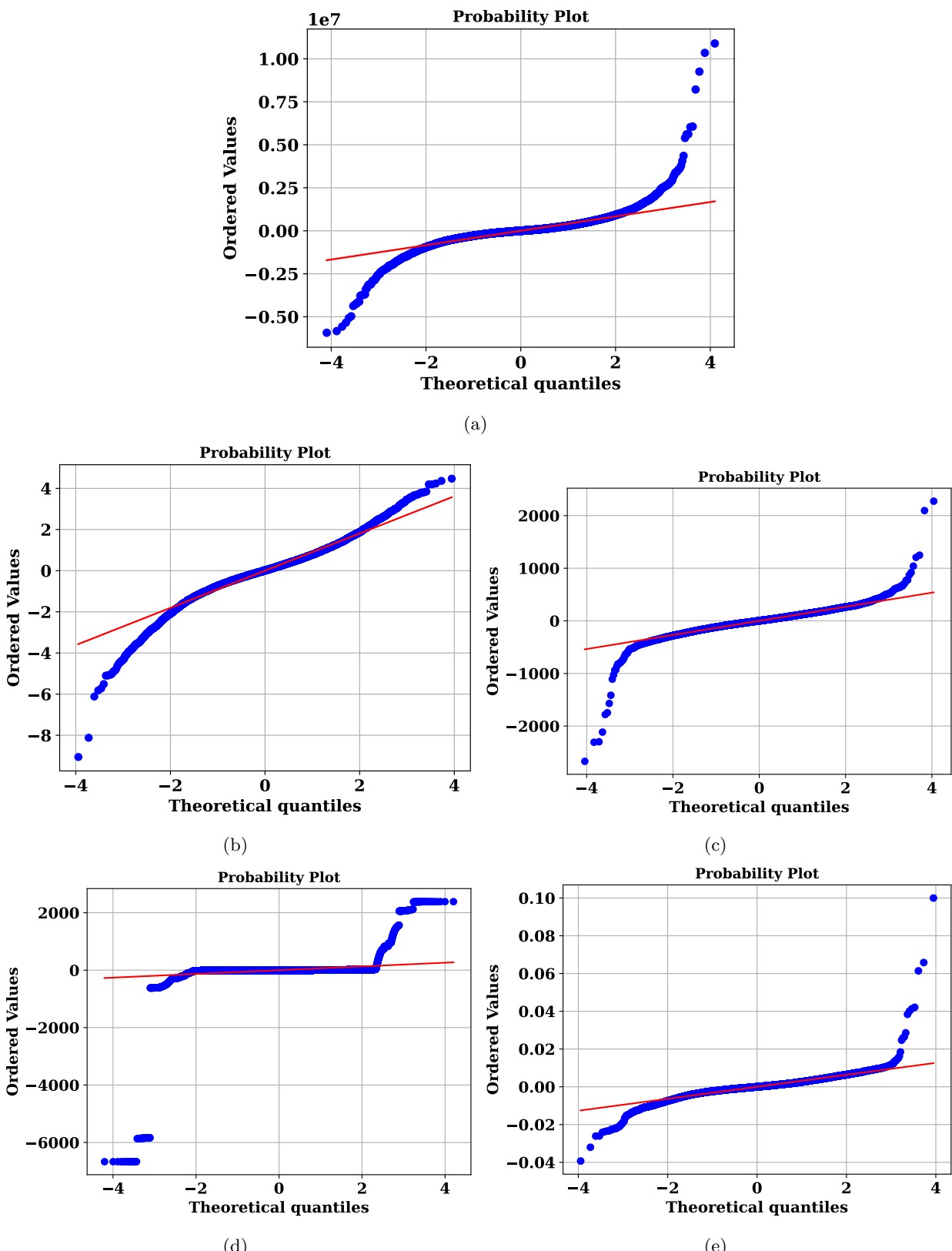

Figure 10: Autocorrelation of time series: (a) Network, (b) ETTh1, (c) Electricity, (d) Weather, (e) Traffic.

Table 8: Number of features in different datasets.

| Dataset | No. of features |
|---------|-----------------|
| **Network** | **47** |
| ETTh1 | 8 |
| Electricity | 322 |
| Weather | 22 |
| Traffic | 863 |

Table 9: Features used in multivariate setting.

| Feature | Description |
|---------|-------------|
| CQI | Channel Quality Indicator |
| MCS | Modulation and Coding Scheme |
| pkt ok | Number of packets sent |
| pkt nok | Number of packets dropped |
| id_ue | Number of ue's connected in the BS |
| pusch_sinr | Noise ratio interference in the Physical Uplink Shared Channel |
| pucch_sinr | Noise ratio interference in the Physical Uplink Control Channel |
| pusch_rssi | Signal strength in the Physical Uplink Shared Channel |
| pucch_rssi | Signal strength in the Physical Uplink Control Channel |
| pucch samples | Number of PUCCH samples |

Table 10: Performance of benchmarked models using ten features.

| Model | Multivariate | |
|-------|------|-----|
| | RMSE | MAE |
| XGB | 0.0347 | 0.0234 |
| ARF | **0.0273** | **0.0155** |
| Naive | 0.0417 | 0.0239 |
| TTM (Zero-shot) | 0.0359 | 0.0230 |
| TTM (Fine-tuning) | 0.0358 | 0.0228 |
| Chronos (Zero-shot) | 0.0285 | 0.0181 |
| Chronos (Fine-tuning) | 0.0280 | 0.0176 |

## A.4 Ablation Study

### A.4.1 Number of features

Table 8 summarizes the number of features in each dataset. Our network data contains 47 features in total, providing enough features for multivariate setting. This ensures that our dataset is well-suited for training TSFMs, similar to existing pre-trained datasets. In our main experiments, we used a subset of four important features from our network dataset as mentioned in Table 2. We extended our analysis to include ten features in total, as shown in Table 9, and evaluated the performance of the benchmarked models in this multivariate

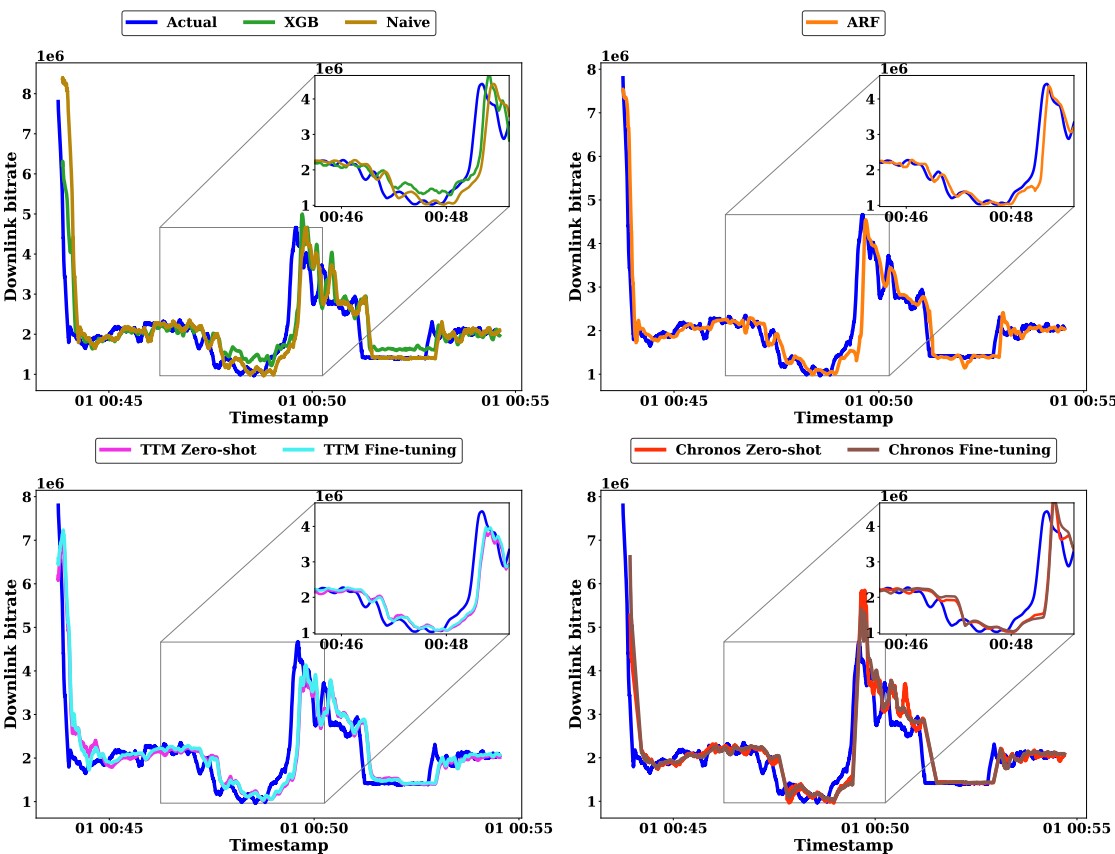

Figure 11: Actual v.s. Predicted bitrate values using ten features.

setting. Table 10 shows that ARF outperforms all the other benchmarked models in this multivariate setting as well. The performance of these models is more clearly reflected in Fig.11. We observe that ARF follows the curve/trend of the bitrate much better than other benchmarked models. All models were evaluated using their default hyper-parameters.

Table 11: Performance metrics of benchmarked models the filtered subset defined by the *train* mobility pattern within the *Dos-Hulk-C* traffic class.

|                    | Univariate |        | Multivariate |        |
| ------------------ | ---------- | ------ | ------------ | ------ |
| Model              | RMSE       | MAE    | RMSE         | MAE    |
| XGB                | 0.1440     | 0.1087 | 0.1440       | 0.1087 |
| ARF                | 0.1728     | 0.1125 | **0.0968**   | **0.0634** |
| Naive              | 0.1309     | 0.0932 | 0.1309       | 0.0932 |
| TTM (Zero-shot)    | **0.1279** | **0.0922** | 0.1279   | 0.0922 |

### A.4.2 Filtered data evaluation

In this section, we evaluate the performance of the benchmarked models on data distributions that differ from those presented in Section 3.2, 5.2 and 6. This analysis also demonstrates the potential of the dataset for transfer learning use cases. By training models on one set of mobility patterns and traffic classes and evaluating them on a different set, we assess how well knowledge learned in one context generalizes to another.

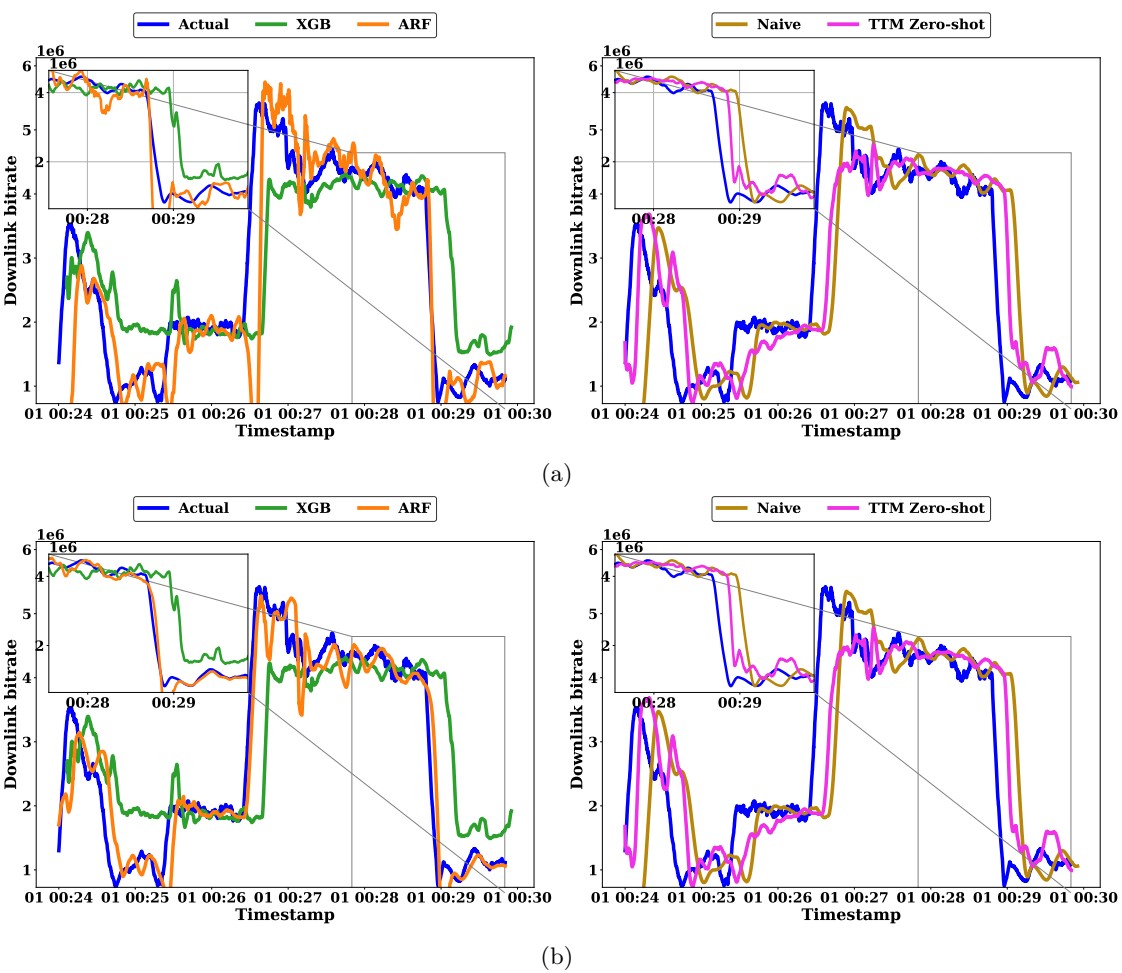

Figure 12: Actual v.s. Predicted bitrate values for the filtered subset: (a) Univariate, (b) Multivariate.

Table 12: Performance metrics of benchmarked models across additional filtered subsets with different mobility patterns and traffic classes.

| Mobility Pattern | Multivariate | | | | | | | |
| | Traffic Class : YouTube | | | | Traffic Class : Portscan | | | |
| | Model | | | | | | | |
| | ARF | | TTM (Zero-shot) | | ARF | | TTM (Zero-shot) | |
| | RMSE | MAE | RMSE | MAE | RMSE | MAE | RMSE | MAE |
| Static | **0.0166** | **0.0123** | 0.0359 | 0.0230 | **0.0858** | **0.0585** | 0.1192 | 0.0904 |
| Pedestrian | **0.0457** | **0.0262** | 0.0765 | 0.0434 | **0.0819** | **0.0555** | 0.1127 | 0.0860 |
| Bus | **0.0790** | **0.0406** | 0.0805 | 0.0481 | **0.0522** | **0.0337** | 0.0945 | 0.0712 |
| Train | **0.0564** | **0.0303** | 0.0623 | 0.0443 | 0.1684 | **0.1180** | **0.1681** | 0.1308 |
| Car | **0.0404** | **0.0217** | 0.0568 | 0.0320 | **0.0819** | **0.0555** | 0.1375 | 0.1037 |

The raw data is filtered based on mobility patterns and traffic generated from *malicious activities*, focusing specifically on the *train* mobility pattern within the *Dos-Hulk-C* traffic class. Table 11 presents the performance of ARF and TTM, evaluated using RMSE and MAE in both univariate and multivariate settings. We

Table 13: Hyper-parameter tuning of the TTM model in the multivariate setting.

| Learning Rate (LR) | RMSE | MAE | Fine-tune Percent | RMSE | MAE | No. of Epochs | RMSE | MAE |
|---|---|---|---|---|---|---|---|---|
| 0.01 | 0.0390 | 0.0249 | 10 | 0.0358 | 0.0227 | 50 | 0.0359 | 0.0227 |
| 0.001 | 0.0387 | 0.0247 | 15 | 0.0365 | 0.0226 | 80 | 0.0359 | 0.0227 |
| 0.00001 | 0.0359 | 0.0227 | 25 | 0.0366 | 0.0227 | 100 | 0.0359 | 0.0227 |
| 0.000001 | 0.0359 | 0.0229 | 30 | 0.0367 | 0.0226 | | | |

Table 14: Hyper-parameter tuning of Lag-Llama model.

| Context Length | RMSE | MAE | Batch Size | RMSE | MAE |
|---|---|---|---|---|---|
| 15 | 0.0350 | 0.0231 | 16 | 0.0330 | 0.0227 |
| 25 | 0.0324 | 0.0217 | 32 | 0.0314 | 0.0218 |
| 35 | 0.0327 | 0.0221 | 128 | 0.0332 | 0.0221 |

Table 15: PatchTST hyper-parameter tuning configurations in the multivariate setting. Case 1 varies the number of attention heads, Case 2 varies the number of encoder layers, Cases 3-5 examine the effect of increasing model width, i.e., $(d_{\mathrm{model}}, d_{ff}) \in \{(128, 256), (256, 512), (512, 1024)\}$.

| Case | $d_{\mathrm{model}}$ | $d_{ff}$ | Heads | Layers | RMSE | MAE |
|---|---|---|---|---|---|---|
| 1 | 64 | 128 | 2 | 3 | 0.0321 | 0.0206 |
| | | | 4 | | **0.0319** | **0.0205** |
| 2 | 64 | 128 | 8 | 2 | 0.0320 | 0.0207 |
| | | | | 4 | 0.0323 | 0.0207 |
| 3 | 128 | 256 | | | 0.0324 | 0.0208 |
| 4 | 256 | 512 | 8 | 3 | 0.0324 | 0.0209 |
| 5 | 512 | 1024 | | | 0.0322 | 0.0209 |

Table 16: iTransformer hyper-parameter tuning configurations in the multivariate setting. Case 1 varies the number of attention heads, Case 2 varies the number of encoder layers, Cases 3-5 examine the effect of increasing model width, i.e., $(d_{\mathrm{model}}, d_{ff}) \in \{(128, 256), (256, 512), (512, 1024)\}$.

| Case | $d_{\mathrm{model}}$ | $d_{ff}$ | Heads | Layers | RMSE | MAE |
|---|---|---|---|---|---|---|
| 1 | 64 | 128 | 2 | 3 | 0.0326 | 0.0205 |
| | | | 4 | | 0.0319 | 0.0206 |
| 2 | 64 | 128 | 8 | 2 | 0.0321 | 0.0206 |
| | | | | 4 | 0.0324 | 0.0209 |
| 3 | 128 | 256 | | | 0.0321 | 0.0207 |
| 4 | 256 | 512 | 8 | 3 | 0.0318 | 0.0207 |
| 5 | 512 | 1024 | | | **0.0317** | **0.0206** |

include TTM in this analysis due to its computational efficiency. For this filtered dataset, TTM outperforms ARF in the univariate setting, while ARF achieves better performance in the multivariate setting. Fig.12 further illustrates how these models capture the trend of the bitrate.

We further extend this evaluation to additional filtered subsets, each representing distinct combinations of mobility patterns and traffic classes. In this case, we restrict the analysis to the multivariate setting. Table 12 shows that TTM consistently performs worse than ARF across most traffic labels. These findings reinforce the stronger generalization capability of ARF in the multivariate setting.

Table 17: Hyper-parameter optimization results for XGBoost in the multivariate setting. Each cell reports RMSE / MAE.

| n_estimators | Depth | LR = 0.1 | LR = 0.3 | LR = 0.5 |
|---|---|---|---|---|
| 50 | 2 | 0.0342 / 0.0232 | 0.0346 / 0.0231 | 0.0350 / 0.0233 |
| | 4 | **0.0339 / 0.0225** | 0.0347 / 0.0228 | 0.0356 / 0.0233 |
| | 6 | 0.0341 / 0.0225 | 0.0352 / 0.0230 | 0.0366 / 0.0238 |
| | 8 | 0.0341 / 0.0225 | 0.0356 / 0.0234 | 0.0381 / 0.0248 |
| 100 | 2 | 0.0341 / 0.0228 | 0.0346 / 0.0230 | 0.0350 / 0.0232 |
| | 4 | 0.0341 / 0.0225 | 0.0350 / 0.0229 | 0.0374 / 0.0244 |
| | 6 | 0.0343 / 0.0225 | 0.0356 / 0.0232 | 0.0356 / 0.0232 |
| | 8 | 0.0344 / 0.0225 | 0.0364 / 0.0240 | 0.0393 / 0.0261 |
| 200 | 2 | 0.0341 / 0.0227 | 0.0347 / 0.0229 | 0.0350 / 0.0231 |
| | 4 | 0.0344 / 0.0225 | 0.0352 / 0.0231 | 0.0364 / 0.0238 |
| | 6 | 0.0344 / 0.0226 | 0.0363 / 0.0238 | 0.0386 / 0.0255 |
| | 8 | 0.0347 / 0.0227 | 0.0363 / 0.0238 | 0.0401 / 0.0269 |
| 500 | 2 | 0.0343 / 0.0226 | 0.0375 / 0.0249 | 0.0350 / 0.0231 |
| | 4 | 0.0346 / 0.0227 | 0.0359 / 0.0236 | 0.0377 / 0.0247 |
| | 6 | 0.0343 / 0.0226 | 0.0375 / 0.0249 | 0.0350 / 0.0231 |
| | 8 | 0.0353 / 0.0233 | 0.0376 / 0.0252 | 0.0402 / 0.0270 |

### A.4.3 TSFMs hyper-parameter tuning

In this section, we evaluate the performance of TTM and Lag-Llama models under different hyper-parameter settings. Table 13 summarizes the results of hyper-parameter tuning for the TTM model in the multivariate setting. It presents the performance of the TTM model under various learning rates, fine-tuning percentages, and number of epochs. We first evaluated the different learning rates, observing that a learning rate of 0.00001 achieves the lowest errors. Using this optimal learning rate (0.00001), we further experimented with different fine-tuning percentages, observing that fine-tuning 10% of training data results in lower RMSE and while the MAE remains largely similar across all fine-tuning percentages. We also tuned the number of training epochs using this learning rate, but found that increasing the number of epochs did not significantly change either the RMSE or MAE, indicating that the performance is largely insensitive to the number of epochs beyond the default setting. TTM provides a built-in learning rate finder, which we used to determine the optimal learning rate for the main results presented in Table 4. Using the learning rate finder algorithm, we obtained an optimal learning rate of 0.0011 for the main results in Table 4, resulting in an RMSE of 0.0391 and an MAE of 0.0249. This shows that tuning the learning rate can noticeably improve the performance of the TTM model after fine-tuning. Nevertheless, ARF continues to outperform TTM.

Further, Table 14 presents the hyper-parameter tuning results for the Lag-Llama model in the univariate setting. We first evaluated different context lengths and observed that a context length of 25 performs better than context length 15. For the main results in Table 4, we used a context length of 5 to ensure a fair comparison with the shallow models, which operate under the same input length constraint. With this context length of 5, Lag-Llama achieves RMSE of 0.0474 and MAE of 0.0268. After selecting the best-

Table 18: Hyperparameter optimization results for Random Forest in the multivariate setting. Each cell reports RMSE / MAE.

| n_estimators | max_depth | max_features = sqrt | max_features = 1 |
|---|---|---|---|
| 50 | None | 0.0342 / 0.0226 | 0.0342 / 0.0228 |
| | 2 | 0.0355 / 0.0259 | 0.0403 / 0.0309 |
| | 4 | 0.0347 / 0.0244 | 0.0355 / 0.0257 |
| 100 | None | 0.0339 / 0.0223 | 0.0342 / 0.0226 |
| | 2 | 0.0355 / 0.0259 | 0.0408 / 0.0314 |
| | 4 | 0.0346 / 0.0244 | 0.0358 / 0.0261 |
| 200 | None | 0.0337 / 0.0222 | 0.0337 / 0.0224 |
| | 2 | 0.0355 / 0.0259 | 0.0399 / 0.0304 |
| | 4 | 0.0346 / 0.0244 | 0.0356 / 0.0260 |
| 400 | None | **0.0336 / 0.0222** | 0.0336 / 0.0223 |
| | 2 | 0.0354 / 0.0258 | 0.0399 / 0.0305 |
| | 4 | 0.0346 / 0.0244 | 0.0355 / 0.0258 |

Table 19: Performance comparison of selected benchmarked models across different prediction horizons in the multivariate setting.

| Model | Prediction Horizon = 192 | | Prediction Horizon = 336 | | Prediction Horizon = 720 | |
|---|---|---|---|---|---|---|
| | RMSE | MAE | RMSE | MAE | RMSE | MAE |
| XGB | 0.0399 | 0.0268 | 0.0445 | 0.0313 | 0.0579 | 0.0436 |
| ARF | **0.0204** | **0.0134** | **0.0187** | **0.0140** | **0.0188** | **0.0138** |
| Naive | 0.0533 | 0.0296 | 0.0623 | 0.0366 | 0.0742 | 0.0506 |
| OLR | 0.0622 | 0.0353 | 0.0681 | 0.0398 | 0.0774 | 0.0488 |
| PatchTST | 0.0357 | 0.0228 | 0.0400 | 0.02584 | 0.0494 | 0.0337 |
| iTransformer | 0.0355 | 0.0226 | 0.0403 | 0.0254 | 0.0496 | 0.0327 |
| TTM (Zero-shot) | 0.0425 | 0.0269 | 0.0499 | 0.0318 | 0.0575 | 0.0403 |
| Chronos (Zero-shot) | 0.0255 | 0.0181 | 0.0255 | 0.0183 | 0.0256 | 0.0194 |

performing context length (25), we then experimented with different batch sizes. As shown in the table, a batch size of 32 provides slightly better performance compared to the other batch sizes.

Similarly, we performed additional experiments in a univariate setting with the zero-shot Chronos-base model, which has 200M parameters, whereas the main results in Table 4 use Chronos-small with 46M parameters. Although Chronos-base achieves slightly lower errors (RMSE: 0.0294, MAE: 0.0179), the improvement over Chronos-small (RMSE: 0.0313, MAE: 0.0185) appears modest despite a more than fourfold increase in model size. Moreover, ARF continues to outperform Chronos in terms of RMSE, indicating that the performance gap is not solely due to model scale.

### A.4.4 Transformer-based deep learning models hyper-parameter tuning

In this section, we evaluate the performance of PatchTST and iTransformer under different hyperparameter configurations in the multivariate setting. Tables 15 and 16 present the performance of PatchTST and

iTransformer under different numbers of attention heads, encoder layers, embedding dimensions ($d_{\mathrm{model}}$), and feed-forward dimensions ($d_{ff}$), respectively.

The hyper-parameter tuning results show that both models are sensitive to hyper-parameter choices, although the extent of the effect differs across architectures. For PatchTST, Case 1 shows that increasing the number of attention heads from 2 to 4 while keeping $d_{\mathrm{model}} = 64$, $d_{ff} = 128$, and the number of encoder layers fixed at 3 leads to a small but consistent improvement, reducing RMSE from 0.0321 to 0.0319 and MAE from 0.0206 to 0.0205. This shows that a moderate increase in attention granularity is beneficial under the compact configuration. In contrast, Case 2 shows that varying the number of encoder layers does not improve performance; using 2 layers results in RMSE = 0.0320 and MAE = 0.0207, while increasing to 4 layers degrades performance further to RMSE = 0.0323 and MAE = 0.0207. Cases 3 to 5 further show that increasing model width does not benefit PatchTST in this setting. When $(d_{\mathrm{model}}, d_{ff})$ is increased from $(64, 128)$ to $(128, 256)$, $(256, 512)$, and $(512, 1024)$ while keeping the number of heads and encoder layers fixed at 8 and 3, respectively, performance becomes slightly worse, with RMSE ranging from 0.0322 to 0.0324 and MAE ranging from 0.0208 to 0.0209. Overall, the best PatchTST result is obtained with a relatively compact configuration, namely $d_{\mathrm{model}} = 64$, $d_{ff} = 128$, 4 attention heads, and 3 encoder layers.

For iTransformer, the tuning results show a different pattern. In Case 1, increasing the number of attention heads from 2 to 4 substantially improves RMSE, reducing it from 0.0326 to 0.0319, while MAE remains nearly unchanged (0.0205 to 0.0206). This indicates that a higher number of attention heads is beneficial for iTransformer under the compact setting, particularly in terms of reducing large forecasting deviations captured by RMSE. Similar to PatchTST, increasing the number of encoder layers in Case 2 does not improve performance. With 2 layers, iTransformer achieves RMSE = 0.0321 and MAE = 0.0206, whereas increasing to 4 layers results in worse performance (RMSE = 0.0324, MAE = 0.0209). However, unlike PatchTST, iTransformer benefits from increasing model width. In Cases 3 to 5, progressively increasing $(d_{\mathrm{model}}, d_{ff})$ from $(128, 256)$ to $(256, 512)$ and finally $(512, 1024)$ results in gradual improvements in RMSE, from 0.0321 to 0.0318 and finally 0.0317, while MAE remains comparatively stable around 0.0206-0.0207. The best overall iTransformer configuration is therefore obtained with $d_{\mathrm{model}} = 512$, $d_{ff} = 1024$, 8 heads, and 3 encoder layers.

Nevertheless, even after tuning these key hyper-parameters, ARF continues to outperform PatchTST and iTransformer.

### A.4.5   Shallow models hyper-parameter tuning

In this section, we evaluate the performance of shallow models, specifically, XGBoost and Random Forest under different hyper-parameter configurations in the multivariate setting. Table 17 presents the hyperparameter optimization results for the XGBoost model across different values of the number of estimators, maximum tree depth, and learning rate (LR). The results show that a lower learning rate of 0.1 consistently produced the best predictive performance across most configurations, while higher learning rates of 0.3 and 0.5 led to increased RMSE and MAE. The best overall performance was obtained with 50 estimators, a maximum depth of 4, and a learning rate of 0.1, achieving an RMSE of 0.0339 and an MAE of 0.0225. The default XGBoost settings are max_depth = 6, learning_rate = 0.3, and n_estimators = 100. Compared with these default values, the best-performing configuration used a smaller number of estimators, a shallower tree structure, and a lower learning rate. Increasing the number of estimators beyond 50 did not lead to further improvement and, in several cases, slightly degraded the performance.

Similarly, Table 18 presents the hyper-parameter optimization results for the Random Forest (RF) model across different values of the number of estimators, maximum tree depth, and the number of features considered at each split. The best overall performance was obtained with 400 estimators, max_depth = None, and max_features = `sqrt`, achieving an RMSE of 0.0336 and an MAE of 0.0222. The default RF settings are max_depth = None, max_features = 1, and n_estimators = 100. The performance improved gradually as the number of estimators increased from 50 to 400, although the gains became smaller at higher values.

Nevertheless, even after tuning these key hyper-parameters, ARF continues to outperform XGBoost and RF.

### A.4.6  Performance on multiple prediction horizons

In this section, we evaluate the performance of selected benchmarked models, chosen based on their computational efficiency, across standard prediction horizons of {192, 336, 720} in the multivariate setting. As shown in Table 19, ARF consistently achieves the best performance across all horizons, with the lowest RMSE and MAE values. While the performance of most models degrades as the prediction horizon increases, ARF remains stable with only minor changes in error. PatchTST and iTransformer perform well at shorter horizons and outperform traditional baselines like XGB, OLR, and the Naive model, however, their performance drops at longer horizons. Among the zero-shot models, Chronos remains stable across all horizons and performs better than all models except ARF. In contrast, TTM's performance worsens with longer horizons, similar to traditional methods.

