# OpenReview forum: "Bridging the High-Frequency Data Gap: A Millisecond-Resolution Network Dataset for Advancing Time Series Foundation Models"
_TMLR — Decision pending for TMLR_

### Review · Reviewer_a7xx · 2026-06-02

**Summary Of Contributions:**

The overall perspective of this work is quite interesting. This paper introduces a millisecond-resolution 5G/O-RAN wireless-network time series dataset and uses it to evaluate several shallow forecasting models and time series foundation models (TSFMs). The dataset contains high-frequency network performance measurements such as downlink bitrate, CQI, MCS, packet statistics, and signal-related indicators, and is motivated as a missing domain and frequency regime for TSFM pre-training and evaluation.

The main empirical result is that Adaptive Random Forest (ARF) outperforms RF, XGBoost, Online LR, naive forecasting, TinyTimeMixer, Chronos, and Lag-Llama on the evaluated bitrate forecasting tasks, especially in the multivariate setting. The paper also shows that the selected network traces are highly non-stationary, weakly seasonal, heavy-tailed, and substantially different from common time-series benchmark datasets.

The topic is timely and potentially valuable. A high-frequency wireless-network benchmark could be useful for studying TSFM generalization. The supplementary source-code folder provides notebooks for the main model families, which is helpful. However, the current submission still needs stronger documentation and reproducibility support. In particular, the dataset files or access instructions, schema, license, preprocessing pipeline, and exact reproduction instructions are not clearly specified. The main experiments are also based on a filtered subset of the dataset, so some conclusions about TSFMs and the full dataset are broader than the evidence currently supports.

**Audience:**

Yes

**Audience Explanation:**

Yes. TMLR readers interested in time series foundation models, out-of-domain generalization, streaming forecasting, and ML for wireless networks would likely find the dataset and negative TSFM results interesting. The high-frequency, noisy, non-stationary wireless setting is underrepresented in current TSFM benchmarks.

The work would become much more useful if the authors provide the dataset or clear access instructions, together with a cleaner reproducibility guide for the supplied notebooks.

**Broader Impact Concerns:**

The broader impact statement is currently too generic. Since the dataset includes wireless telemetry and malicious traffic labels, the paper should discuss anonymization, privacy, data sensitivity, and possible misuse. I do not see an issue that necessarily prevents publication, but this section should be made more specific before acceptance.

**Claims And Evidence:**

No

**Claims Explanation:**

The paper provides useful initial evidence that the proposed 5G traces are challenging for current TSFMs and that ARF is a strong baseline for this setting. The descriptive analysis and benchmark results support the narrower claim that this data distribution differs from standard low-frequency time-series datasets.

However, the central dataset contribution is not yet sufficiently verifiable. I found supplementary notebooks for several models, but they depend on files such as 5G_millisecond.csv and 5G_ablation.csv, and I could not find clear dataset access instructions, license, schema, or a complete run guide. For a dataset paper, these are essential.

The experimental claims are also somewhat too broad. The main benchmark focuses on a specific filtered subset, while the paper motivates a much broader dataset with multiple mobility patterns and traffic classes. Some appendix experiments help, but they do not fully establish broad TSFM failure across the whole dataset. In addition, the fairness of tuning across ARF, shallow baselines, and TSFMs should be clarified, and the use of future covariates in the multivariate Chronos setup needs justification.

**Requested Changes:**

Provide a clear dataset availability statement, including access instructions, license, format, and any restrictions. The source notebooks are useful, but the referenced CSV files are not clearly provided or documented.

Add complete dataset documentation: feature schema, units, number of UEs/traces, duration, labels, missing-value handling, filtering criteria, and preprocessing steps.

Improve the reproducibility package. The supplementary notebooks cover many experiments, but the authors should add a concise run guide, dependency versions beyond the minimal conda files, expected inputs/outputs, and commands or scripts to reproduce the main tables.

Clarify the multivariate forecasting setup, especially whether future covariates such as CQI, MCS, or packet statistics are assumed available at prediction time.

Temper broad conclusions about TSFMs, or expand the main experiments to cover more mobility patterns, traffic classes, and TSFM configurations.

Strengthen the broader impact discussion, especially privacy/anonymization and potential misuse of wireless telemetry or malicious-traffic labels.

Correct the MAE formula in Appendix A.2, which appears to omit the absolute value.

Suggested improvements:

Move a compact version of the appendix filtered-data experiments into the main paper.

Report results across multiple prediction horizons, not only the main 96-step setting.

Clarify normalization and metric interpretation, including whether RMSE/MAE are in normalized units or physical bitrate units.

Improve presentation polish: some figures/tables are hard to read, and there are minor grammar and section-reference issues.

---

> ### Author Response · Authors · 2026-06-17
>
> Thank you for your review comments and feedback on our work. Please find below the updated changes in the revised paper.
>
> **Requested Changes**
> > RC1, RC2, RC3: We have revised the supplementary material to improve both dataset documentation and reproducibility. Specifically, we now include a dataset availability statement with access instructions, format, and license as well as expanded dataset documentation covering the feature schema, units, number of UEs/traces, duration, labels, missing-value handling, filtering criteria, and pre-processing steps. In addition, we have added runnable scripts to reproduce Table 4 and a concise guide describing dependencies, expected inputs/outputs, and execution steps. We are currently extending the script-based reproducibility support to other main results.
>
> > RC4: We have clarified the multivariate forecasting setup (Section 4.3; Post-Processing). In the multivariate setting, the covariates CQI, MCS, pkt_ok, and pkt_nok are treated as synchronously observed features rather than future unknowns. This assumption is not an idealization: in an operational O-RAN deployment, the Distributed Unit (DU) reports these Physical and Medium Access Control (PHY/MAC) layer performance measurements to the near-Real-Time RAN Intelligent Controller (near-RT RIC) via the E2 interface at configurable periodicities in the sub-second range, consistent with the millisecond resolution of our dataset. Consequently, at each prediction timestep t, all covariate values corresponding to that timestep are already observed quantities delivered through standard O-RAN measurement reporting, and no future covariate leakage occurs in our experimental setup.
>
> > RC5: We expanded the evaluation scope by assessing the benchmarked models on a newly filtered data subset. The raw data were filtered based on mobility patterns and traffic generated from benign applications, with a particular focus on the train mobility pattern within the Web Browsing traffic class. Additional evaluations across different mobility patterns and traffic classes are presented in Appendix (Section A.4.2). We have added a new section (Section 6) to the main paper to present these results. We validated that, for this filtered subset as well, ARF consistently outperforms other shallow models, transformer-based deep learning models, and TSFMs in both univariate and multivariate settings.
>
> **Table: Performance metrics on the filtered subset (train mobility, Web Browsing traffic).**
>
> | Model                      | Univariate RMSE | Univariate MAE | Multivariate RMSE | Multivariate MAE |
> |---------------------------|----------|----------|------------|------------|
> | RF                        | 0.0751   | 0.0542   | 0.0740     | 0.0536     |
> | XGB                       | 0.0774   | 0.0536   | 0.0758     | 0.0533     |
> | ARF                       | **0.0605** | **0.0366** | **0.0459** | **0.0230** |
> | Naive                     | 0.0727   | 0.0476   | 0.0727     | 0.0476     |
> | OLR                       | 0.0720   | 0.0470   | 0.0723     | 0.0468     |
> | PatchTST                  | 0.0705   | 0.0468   | 0.0739     | 0.0484     |
> | iTransformer              | 0.0707   | 0.0453   | 0.0726     | 0.0465     |
> | TTM (Zero-shot)           | 0.0718   | 0.0460   | 0.0718     | 0.0460     |
> | TTM (Fine-tuning)         | 0.0719   | 0.0475   | 0.0720     | 0.0488     |
> | Chronos (Zero-shot)       | 0.0724   | 0.0376   | 0.0886     | 0.0473     |
> | Chronos (Fine-tuning)     | 0.0694   | 0.0389   | 0.0846     | 0.0470     |
> | Lag-Llama (Zero-shot)     | 0.0818   | 0.0508   | -          | -          |
> | Lag-Llama (Fine-tuning)   | 0.0771   | 0.0460   | -          | -          |
>
> We also revised the title of our paper to "msData: A Millisecond-Resolution Network Dataset for Advancing Time Series Foundation Models", and narrowed the scope of our claims, explicitly stating that our findings apply only to a very constrained subset of wireless network behavior, rather than 5G networks in general across the paper in the revised version.

---

> > ### Author Response · Authors · 2026-06-17
> >
> > > RC6: We have updated the broader impact statement. The dataset introduced in this work was collected entirely within a controlled testbed environment using software-defined radios and scripted user equipment, with no participation of real subscribers or transmission of user-generated content. All device identifiers are internal testbed labels assigned to specific hardware units and carry no personally identifiable information. Consequently, the dataset raises no privacy or anonymization concerns of the kind associated with operational network deployments. The inclusion of malicious traffic classes, namely DDoS-Ripper, DoS-Hulk, PortScan, and Slowloris, is intended to support intrusion detection and traffic classification research within the O-RAN security domain. The data released consists exclusively of aggregated PHY and MAC layer performance measurements collected at the base station side, such as CQI, MCS, SINR, and packet delivery statistics. These are network-side observability metrics and do not expose packet content, application payloads, or any information that would provide meaningful operational uplift to a prospective attacker. The value of including these classes lies in enabling models that can identify anomalous RAN behavior from standard KPIs alone, without reliance on deep packet inspection, which is a defensive rather than offensive capability.
> >
> > > RC7: We have updated the MAE formula in Appendix Section A.2.
> >
> > ** Suggested Improvements**
> >
> > > SI1: We have added a new section (Section 6) to the main paper to present the expanded evaluation scope, in which we assess the benchmarked models on a newly filtered data subset.
> >
> > > SI2: We conducted experiments using standard prediction horizons of \{192, 336, 720\} for a subset of the benchmarked models. We have added a new section in the Appendix (Section A.4.6) to present these results.
> >
> > > SI3: We have clarified in Section 4.4 that the reported RMSE and MAE values are presented in normalized form.
> >
> > > SI4: We have improved the quality of the figures and tables and corrected grammar and section-reference issues.
> >
> > We hope this reply answers your questions satisfactorily. We are available for further discussion.

---

### Review · Reviewer_YtjM · 2026-06-02

**Summary Of Contributions:**

This paper focuses on the development of TSFMs and introduces a noval high frequency 5G network dataset for TSFM pretraining and evaluation. Its experiment results show that traditional dynamic models like ARF significantly outperform SOTA TSFMs.

Strengths:
- The dataset effectively bridges the "high-frequency data gap" in the TSFM literature. By introducing millisecond-level wireless network measurements, the paper pushes the boundaries of current benchmarks
- The authors demonstrate high technical competence in designing a fair benchmark. The experimental setup meticulously handles data alignment and rolling evaluation procedures to ensure an "apple-to-apple" comparison between pre-trained foundation models and dynamic shallow learners.

Weaknesses:
- Missing deep learning baselines: In the current baseline design, only traditional shallow models and TSFM models are evaluated. While there is a big family of transformer based time series models that are not foundation models (e.g. PatchTST, SOFTS, etc) Without comparing the performance of TSFM models with non-foundation transformer models on the dataset, it remains unclear whether the poor performance of Chronos or TTM is due to the absence of relevant pre-training data, or if the underlying Transformer architectures fundamentally struggle with the highly volatile, non-stationary, and step-like nature of this specific high-frequency network data.
- Disconnect between Dataset Claims and Evaluation Scope: There is a significant disconnect between the rich diversity claimed in the dataset overview (Section 3.1) and the actual data used for the benchmark as the authors "filter for video streaming traffic" only.
- As outlined in the limitation section, the conclusion from the benchmark study is slightly weakened due to the fact that it relies on default hyper-parameters for several shallow baselines while optimizing ARF and not exhausted all techniques of modern fine-tuning techniques of TSFM models.

**Audience:**

Yes

**Audience Explanation:**

Despite the issues with the experimental design, this paper is highly interesting to the TMLR community, especially for those working on TSFMs.

Dataset wise: There is a growing interesting in pushing TSFMs to work across different domains and temporal frequencies, but most research and existing benchmarks focused on relatively stable or predictable data. The community will agree that introducing high-frequency unstable time series data to the training of TSFMs is a reasonable direction.

Experiment conclusion wise: Even if the experiment needs more work, the finding, that "dynamic, traditional models can outperform modern foundation models in chaotic, non-stationary environments" is very interesting and thought-provoking to the community. If the authors could make the benchmark study more strict, the conclusion will inspire the community of reconsider the robustness of the current architectures of SOTA TSFMs. This discussion will be very valuable.

**Claims And Evidence:**

No

**Claims Explanation:**

I don't think the paper's main claims are fully supported by the evidence provided (which can be improved once the requested changes are done), mainly due to the 2 factors:

1. In section 5.1 the authors claims the failure of TSFMs is due to "absence of relevant pre-training data". However, the conclusion is not strong enough because of the miss of deep learning baseline models in the benchmark study.
   - The authors only compare "traditional shallow models" against "pre-trained foundation models." They completely miss the middle ground: standard Transformer-based time series models (like PatchTST or SOFTS) trained from scratch on this specific dataset. As I mentioned before, without it, it remains unclear whether the poor performance of Chronos or TTM is due to the absence of relevant pre-training data, or if the underlying Transformer architectures fundamentally struggle with the highly volatile, non-stationary, and step-like nature of this specific high-frequency network data.

2. The paper title claims it "bridge the high-frequency data gap" for TSFMs, and emphasized the inclusion of different mobility patterns like "car, bus, and train". However, in the experiment study, the authors restrict the evaluation to a subset that "filter for video streaming traffic" in "static" mobility scenarios. This creates a significant gap between the stated contribution and the actual evidence. Video streaming traffic data is expected to have some characteristics like " step-like shifts", "sharp spikes" while other high-frequency dataset does not necessarily have.

**Requested Changes:**

To strengthen the paper and solidify its contributions, I recommend the following adjustments:
1. (Critical) Include deep-learning/transformer baseline: To isolate whether TSFMs’ poor performance is due to a lack of pre-training data or fundamental architectural limitations, the authors must include at least one or two standard Transformer-based models (e.g., PatchTST or SOFTS) trained from scratch on this specific dataset.
2. (Critical) Expand Evaluation Scope or Reduce the Claim: The benchmark is currently restricted to the "Static" mobility + "Video Streaming" subset.
     - If expanding: Please evaluate the baselines on more diverse slices of the data, such as a highly dynamic mobility pattern (e.g., "Car" or "Train") and at least one other traffic class (e.g., "Web Browsing" or "VoIP").
     - If reducing: If these diverse scenarios are not feasible to include, the authors must reduce the scope of their claim, explicitly stating that their findings apply only to a very constrained subset of wireless network behavior, rather than 5G networks in general.
3. (Optional) Try to apply a consistent hyperparameter optimization process across all shallow models if possible.
4. (Minor) There are some editing errors need to be fixed:
    - In Section 3.2, the paper references a "Section 9" regarding autocorrelation analysis ("...see Section 9..."). However, the paper only contains 7 sections and an appendix. Please verify this citation and correct it, as it is confusing for the reader.
    - In the last paragraph of Section when introducing the structure of the paper, it skipped Section 6.

---

> ### Author Response · Authors · 2026-06-17
>
> Thank you for your review comments and feedback on our work. Please find below the updated changes in the revised paper.
>
> **Requested Changes**
> > RC1: We included transformer-based models trained from scratch on our network dataset, specifically PatchTST and iTransformer, as deep learning baselines for benchmarking. We validated that that ARF still outperforms both of the models. We have updated Table 4, along with Figures 5 and 6, with these results as well. Since these models are trained from scratch, we adopt a lightweight configuration by setting the embedding dimension to (d_model = 64) and the feed-forward dimension to (d_ff = 2 * d_model = 128), while keeping all other hyper-parameters at their default values.
>
> **Table: Performance metrics of benchmarked models (mean ± std).**
>
> | Model        | Univariate RMSE           | Univariate MAE            | Multivariate RMSE         | Multivariate MAE          |
> |--------------|-------------------|--------------------|--------------------|--------------------|
> | ARF          | **0.0270 ± 0.0002** | 0.0189 ± 0.0001    | **0.0175 ± 0.0007** | **0.0130 ± 0.0005** |
> | PatchTST     | 0.0327 ± 0.00044  | 0.0212 ± 0.00035   | 0.0321 ± 0.00029   | 0.0207 ± 0.00009   |
> | iTransformer | 0.0325 ± 0.00013  | 0.0208 ± 0.00019   | 0.0324 ± 0.00015   | 0.0208 ± 0.00006   |
>
>
> We also conducted additional hyper-parameter tuning to further evaluate the performance of these models in a multivariate setting. As shown in tables below, tuning the hyper-parameters noticeably improves the performance for both models; however, ARF continues to outperform both PatchTST and iTransformer.
>
> **Table: PatchTST hyper-parameter tuning (multivariate setting).**
>
> | Case | d_model | d_ff | Heads | Layers | RMSE  | MAE  |
> |------|--------:|-----:|------:|-------:|------:|-----:|
> | 1    | 64      | 128  | 2     | 3      | 0.0321 | 0.0206 |
> | 1    | 64      | 128  | 4     | 3      | **0.0319** | **0.0205** |
> |------|---------|------|-------|--------|--------|-------|
> | 2    | 64      | 128  | 8     | 2      | 0.0320 | 0.0207 |
> | 2    | 64      | 128  | 8     | 4      | 0.0323 | 0.0207 |
> |------|---------|------|-------|--------|--------|-------|
> | 3    | 128     | 256  | 8     | 3      | 0.0324 | 0.0208 |
> | 4    | 256     | 512  | 8     | 3      | 0.0324 | 0.0209 |
> | 5    | 512     | 1024 | 8     | 3      | 0.0322 | 0.0209 |
>
>
> **Table: iTransformer hyper-parameter tuning (multivariate setting).**
>
> | Case | d_model | d_ff | Heads | Layers | RMSE  | MAE  |
> |------|--------:|-----:|------:|-------:|------:|-----:|
> | 1    | 64      | 128  | 2     | 3      | 0.0326 | 0.0205 |
> | 1    | 64      | 128  | 4     | 3      | 0.0319 | 0.0206 |
> |------|---------|------|-------|--------|--------|-------|
> | 2    | 64      | 128  | 8     | 2      | 0.0321 | 0.0206 |
> | 2    | 64      | 128  | 8     | 4      | 0.0324 | 0.0209 |
> |------|---------|------|-------|--------|--------|-------|
> | 3    | 128     | 256  | 8     | 3      | 0.0321 | 0.0207 |
> | 4    | 256     | 512  | 8     | 3      | 0.0318 | 0.0207 |
> | 5    | 512     | 1024 | 8     | 3      | **0.0317** | **0.0206** |

---

> ### Author Response · Authors · 2026-06-17
>
> > RC2: We expanded the evaluation scope by assessing the benchmarked models on a newly filtered data subset. The raw data were filtered based on mobility patterns and traffic generated from benign applications, with a particular focus on the train}mobility pattern within the Web Browsing traffic class. Additional evaluations across different mobility patterns and traffic classes are presented in Appendix (Section A.4.2). We have added a new section (Section 6) to the main paper to present these results. We validated that, for this filtered subset as well, ARF consistently outperforms other shallow models, transformer-based deep learning models, and TSFMs in both univariate and multivariate settings.
>
> **Table: Performance metrics on the filtered subset (train mobility, Web Browsing traffic).**
> | Model                      | Univariate RMSE | Univariate MAE | Multivariate RMSE | Multivariate MAE |
> |---------------------------|----------|----------|------------|------------|
> | RF                        | 0.0751   | 0.0542   | 0.0740     | 0.0536     |
> | XGB                       | 0.0774   | 0.0536   | 0.0758     | 0.0533     |
> | ARF                       | **0.0605** | **0.0366** | **0.0459** | **0.0230** |
> | Naive                     | 0.0727   | 0.0476   | 0.0727     | 0.0476     |
> | OLR                       | 0.0720   | 0.0470   | 0.0723     | 0.0468     |
> | PatchTST                  | 0.0705   | 0.0468   | 0.0739     | 0.0484     |
> | iTransformer              | 0.0707   | 0.0453   | 0.0726     | 0.0465     |
> | TTM (Zero-shot)           | 0.0718   | 0.0460   | 0.0718     | 0.0460     |
> | TTM (Fine-tuning)         | 0.0719   | 0.0475   | 0.0720     | 0.0488     |
> | Chronos (Zero-shot)       | 0.0724   | 0.0376   | 0.0886     | 0.0473     |
> | Chronos (Fine-tuning)     | 0.0694   | 0.0389   | 0.0846     | 0.0470     |
> | Lag-Llama (Zero-shot)     | 0.0818   | 0.0508   | -          | -          |
> | Lag-Llama (Fine-tuning)   | 0.0771   | 0.0460   | -          | -          |
>
> We also revised the title of our paper to "msData: A Millisecond-Resolution Network Dataset for Advancing Time Series Foundation Models", and narrowed the scope of our claims, explicitly stating that our findings apply only to a very constrained subset of wireless network behavior, rather than 5G networks in general across the paper in the revised version.
>
> > Optional Comment: We conducted hyper-parameter optimization for XGB and RF in the multivariate setting, similar to ARF, using the same random search methodology. We have added a new section in the Appendix (Section A.4.5) to present these results.
>
> > Minor Comment: We have corrected the reference issues.
>
> We hope this reply answers your questions satisfactorily. We are available for further discussion.

---

### Review · Reviewer_nwhq · 2026-06-03

**Summary Of Contributions:**

The paper introduces a novel millisecond-resolution time series dataset collected from a real-world 5G Open Radio Access Network (O-RAN) deployment. This dataset aims to address a critical gap in the development of Time Series Foundation Models (TSFMs), which are predominantly trained on low-frequency data (ranging from seconds to years). The authors benchmark several shallow machine learning models (e.g., Random Forest, XGBoost, Adaptive Random Forest) against state-of-the-art TSFMs (e.g., TTM, Chronos, Lag-Llama) on short-term forecasting tasks. The empirical results demonstrate that current TSFMs struggle to generalize to this high-frequency, non-stationary data, consistently underperforming compared to adaptive shallow models like ARF.

Key Strengths:

1. The paper highlights and addresses the lack of high-frequency (millisecond-level) datasets in the current TSFM ecosystem.

2. Providing a real-world dataset from the wireless network domain (5G O-RAN) adds valuable diversity to existing open datasets like energy, finance, and weather.

**Audience:**

Yes

**Audience Explanation:**

The TMLR audience includes many researchers focused on foundation models, time series analysis, and network optimization. Because TSFMs are currently a highly active area of research, understanding their boundaries and limitations, specifically their failure to generalize to high-frequency, highly volatile data without structural periodicity, is of significant interest.

**Broader Impact Concerns:**

None.

**Claims And Evidence:**

Yes

**Claims Explanation:**

The authors thoroughly support their claims with robust empirical evidence. The core claim that TSFMs perform poorly on high-frequency wireless data is clearly backed by quantitative results in Tables 4, 6, and 9. Furthermore, the claim that this dataset's characteristics fundamentally differ from standard benchmarks is supported by clear visualizations and statistical tools, such as the STL decomposition and signal-to-noise ratio analyses (Figs. 4, 7, 8). The ablation studies on temporal resolution and fine-tuning strategies further validate their findings.

**Requested Changes:**

Although the focus is on comparing shallow models to TSFMs, conducting standard Hyperparameter Optimization (HPO) on XGBoost and Random Forest would make the baseline benchmarks more rigorous and unassailable for future comparative studies.

---

> ### Author Response · Authors · 2026-06-17
>
> Thank you for your review comments and feedback on our work. Please find below the updated changes in the revised paper.
>
> > For RC1, we conducted hyper-parameter optimization for XGBoost and Random Forest in the multivariate setting. As shown in the tables below, tuning the parameters improves the performance of both models; however, ARF continues to outperform both XGB and RF. We have added a new section in the Appendix (Section A.4.5) to present these results.
>
> **Table: Hyperparameter optimization results for XGBoost in the multivariate setting (RMSE / MAE).**
>
> | n_estimators | depth | LR=0.1           | LR=0.3           | LR=0.5           |
> |--------------|-------|------------------|------------------|------------------|
> | 50           | 2     | 0.0342 / 0.0232  | 0.0346 / 0.0231  | 0.0350 / 0.0233  |
> | 50           | 4     | **0.0339 / 0.0225** | 0.0347 / 0.0228  | 0.0356 / 0.0233  |
> | 50           | 6    | 0.0341 / 0.0225 | 0.0352 / 0.0230  | 0.0366 / 0.0238  |
> | 50           | 8     | 0.0341 / 0.0225 | 0.0356 / 0.0234  | 0.0381 / 0.0248  |
> |---------------------------------------------|---------------------------------
> | 100          | 2     | 0.0341 / 0.0228  | 0.0346 / 0.0230  | 0.0350 / 0.0232  |
> | 100          | 4     | 0.0341 / 0.0225  | 0.0350 / 0.0229  | 0.0374 / 0.0244  |
> | 100          | 6     | 0.0343 / 0.0225  | 0.0356 / 0.0232  | 0.0356 / 0.0232  |
> | 100          | 8     | 0.0344 / 0.0225  | 0.0364 / 0.0240  | 0.0393 / 0.0261  |
> |---------------------------------------------|---------------------------------
> | 200          | 2     | 0.0341 / 0.0227  | 0.0347 / 0.0229  | 0.0350 / 0.0231  |
> | 200          | 4     | 0.0344 / 0.0225  | 0.0352 / 0.0231  | 0.0364 / 0.0238  |
> | 200          | 6     | 0.0344 / 0.0226  | 0.0363 / 0.0238  | 0.0386 / 0.0255  |
> | 200          | 8     | 0.0347 / 0.0227  | 0.0363 / 0.0238  | 0.0401 / 0.0269  |
> |---------------------------------------------|---------------------------------
> | 500          | 2     | 0.0343 / 0.0226  | 0.0375 / 0.0249  | 0.0350 / 0.0231  |
> | 500          | 4     | 0.0346 / 0.0227  | 0.0359 / 0.0236  | 0.0377 / 0.0247  |
> | 500          | 6     | 0.0343 / 0.0226  | 0.0375 / 0.0249  | 0.0350 / 0.0231  |
> | 500          | 8     | 0.0353 / 0.0233  | 0.0376 / 0.0252  | 0.0402 / 0.0270  |
>
>
>
> **Table: Hyperparameter optimization results for Random Forest in the multivariate setting (RMSE / MAE).**
>
> | n_estimators | max_depth | max_features = sqrt | max_features = 1 |
> |--------------|-----------|---------------------|------------------|
> | 50           | None      | 0.0342 / 0.0226     | 0.0342 / 0.0228  |
> | 50           | 2         | 0.0355 / 0.0259     | 0.0403 / 0.0309  |
> | 50           | 4         | 0.0347 / 0.0244     | 0.0355 / 0.0257  |
> |--------------|-----------|---------------------|------------------|
> | 100          | None      | 0.0339 / 0.0223     | 0.0342 / 0.0226  |
> | 100          | 2         | 0.0355 / 0.0259     | 0.0408 / 0.0314  |
> | 100          | 4         | 0.0346 / 0.0244     | 0.0358 / 0.0261  |
> |--------------|-----------|---------------------|------------------|
> | 200          | None      | 0.0337 / 0.0222     | 0.0337 / 0.0224  |
> | 200          | 2         | 0.0355 / 0.0259     | 0.0399 / 0.0304  |
> | 200          | 4         | 0.0346 / 0.0244     | 0.0356 / 0.0260  |
> |--------------|-----------|---------------------|------------------|
> | 400          | None      | **0.0336 / 0.0222** | 0.0336 / 0.0223  |
> | 400          | 2         | 0.0354 / 0.0258     | 0.0399 / 0.0305  |
> | 400          | 4         | 0.0346 / 0.0244     | 0.0355 / 0.0258  |
>
>
> We hope this reply answers your questions satisfactorily. We are available for further discussion.